# Homeostatic control of an iron repressor in a GI tract resident

Yuanyuan Wang[1†], Yinhe Mao[1,2†], Xiaoqing Chen[1,2], Xinhuang Huang[1], Zhongyi Jiang[3], Kaiyan Yang[1,2], Lixing Tian[4], Tong Jiang[1], Yun Zou[1], Xiaoyuan Ma[4], Chaoyue Xu[1,5], Zili Zhou[1], Xianwei Wu[1,2], Lei Pan[1], Huaping Liang[4*], Lin Zhong[3*], Changbin Chen[1*]

[1]The Center for Microbes, Development and Health, Key Laboratory of Molecular Virology and Immunology, Unit of Pathogenic Fungal Infection & Host Immunity, Institut Pasteur of Shanghai, Chinese Academy of Sciences, Shanghai, China; [2]The University of Chinese Academy of Sciences, Beijing, China; [3]Department of General Surgery, Shanghai General Hospital, Shanghai, China; [4]State Key Laboratory of Trauma, Burns and Combined Injury, Department of Wound Infection and Drug, Army Medical Center (Daping Hospital), Chongqing, China; [5]College of Life Science, Shanghai, China

*For correspondence:
13638356728@163.com (HL);
zhonglin1@medmail.com.cn (LZ);
cbchen@ips.ac.cn (CC)

†These authors contributed equally to this work

**Abstract** The transition metal iron plays a crucial role in living cells. However, high levels of iron are potentially toxic through the production of reactive oxygen species (ROS), serving as a deterrent to the commensal fungus *Candida albicans* for colonization in the iron-rich gastrointestinal tract. We observe that the mutant lacking an iron-responsive transcription factor Hap43 is hyper-fit for colonization in murine gut. We demonstrate that high iron specifically triggers multiple post-translational modifications and proteasomal degradation of Hap43, a vital process guaranteeing the precision of intestinal ROS detoxification. Reduced levels of Hap43 de-repress the expression of antioxidant genes and therefore alleviate the deleterious ROS derived from iron metabolism. Our data reveal that Hap43 functions as a negative regulator for oxidative stress adaptation of *C. albicans* to gut colonization and thereby provide a new insight into understanding the interplay between iron homeostasis and fungal commensalism.

## Editor's evaluation

This important study reveals a novel mechanism that links iron stress to anti-oxidant protection for the gut commensal and opportunistic pathogen *Candida albicans* in colonization of the gastrointestinal tract. Through a series of convincing experiments, the authors show that phosphorylation and degradation of Hap43, a well-established regulator of *Candida albicans* iron homeostasis, underlies the interaction between this gut commensal and the mammalian gut. The work will be of interest to microbiologists working on microbiota and microbial commensalism.

## Introduction

Iron is an essential element required for the viability of virtually all organisms (*Andrews, 2008*). This transition metal acts as an enzyme cofactor, predominantly in electron transfer and catalysis, and therefore contributes to numerous metabolic processes, in particular energy generation, oxygen transport,

and DNA synthesis. However, excess of iron is toxic and potentially fatal, primarily because reactive oxygen species (ROS), including hydroxyl radicals (OH·), superoxide (O2·), and $H_2O_2$, are generated through the iron-catalyzed Haber-Weiss/Fenton reaction and causes cell damage and death (*Pierre et al., 2002*). The mammalian gut is considered as an iron-rich environment where large amounts of dietary iron (e.g. ~0.27 mM per day in humans) are regularly present in the colonic lumen and remain unabsorbed (*Miret et al., 2003*). Interestingly, previous studies have found that the concentration of iron in feces of British adults regularly consumed a standard Western diet and of weaning infants fed with complementary foods could reached to an average value of 100 μg/g wet weight feces (~1.8 mM), and this level is actually much higher than the minimal concentration (~0.4 mM) required for intestinal bacterial species (*Lund et al., 1998*; *Lund et al., 1999*; *Pizarro et al., 1987*). Hence, it is very likely that the increased level of unabsorbed iron would aggravate the status of oxidative stress in the gastrointestinal (GI) tract, providing a detrimental impact on the growth of resident microbial commensals. Indeed, excessive ROS levels in this iron-replete niche were found to enhance cellular toxicity, reflected by oxidative damage to proteins, lipids, and DNA, and therefore restrict the growth and proliferation of colonized microorganisms (*Dixon and Stockwell, 2014*; *Schieber and Chandel, 2014*).

The potent redox capability of iron requires that microbes carefully respond to and regulate environmental iron levels and distribution (*Barber and Elde, 2015*). Several examples of iron detoxification have been described in bacterial species. For example, both *Escherichia coli* and *Salmonella typhimurium* developed effective iron efflux systems to decrease intracellular accumulation of free iron and prevent iron stress (*Frawley et al., 2013*; *Grass et al., 2005*). Similarly, eukaryotic microbes like fungi also decreased the labile iron pool to prevent formation of deleterious hydroxyl radicals through the vacuolar and siderophore-mediated iron storage (*Gupta and Outten, 2020*; *Singh et al., 2007*). However, studies investigating mechanisms employed by gut microbes to detoxify iron-mediated ROS production are relatively limited.

*Candida albicans* is a major opportunistic fungal pathogen of humans, capable of causing mucosal diseases with substantial morbidity and life-threatening bloodstream infections in immunocompromised individuals (*da Silva Dantas et al., 2016*; *Noble and Johnson, 2007*). Importantly, this fungus is also the most prevalent fungal species of the human microbiota and acts as a commensal to effectively colonize many host niches, particularly the GI tract (*Kumamoto et al., 2020*). Our previous studies have demonstrated that the acquisition and storage of iron in *C. albicans* were effectively regulated by a complex and effective regulatory circuit, which consists of three iron-responsive transcription regulators (Sfu1, Sef1, and Hap43) and plays reciprocal roles in regulating *C. albicans* commensalism and pathogenesis (*Chen et al., 2011*). Specifically, the GATA family transcription factor Sfu1 was found to downregulate the expression of iron acquisition genes, prevent toxicity in iron-replete conditions, and contribute to intestinal commensalism of *C. albicans*. Sef1 acts as the central iron regulator for the expression of iron uptake genes in low-iron conditions and surprisingly plays a dual role in regulating both intestinal commensalism and virulence (*Chen et al., 2011*). The CCAAT-binding repressor Hap43 transcriptionally represses Sfu1 that therefore de-represses iron acquisition gene expression in iron-limited conditions, and evidence has shown that mutant lacking *HAP43* exhibits a defect in virulence, supporting its role in pathogenicity (*Chen et al., 2011*; *Hsu et al., 2011*; *Singh et al., 2011*). Although the iron homeostasis regulatory circuit, as shown above, is essentially required for both commensalism and pathogenesis, the exact regulatory mode for each of the three factors and its application in driving the transition of *C. albicans* commensalism and pathogenicity remains largely unsolved.

The CCAAT-binding factor is a highly conserved heteromeric transcription factor that specifically binds to the 5′-CCAAT-3′ consensus elements within the promoters of numerous eukaryotic genes (*Kato, 2005*). In mammals, three subunits, including NF-YA, NF-YB, and NF-YC, form an evolutionarily conserved nuclear factor Y (NY-F) complex that exhibits the DNA-binding capacity to the CCAAT box and plays a vital role in transcriptional regulation of genes involved in proliferation and apoptosis, cancer and tumor, stress responses, growth, and development (*Dorn et al., 1987*). A similar NF-Y-like (HAP) complex also exists in fungi like the budding yeast *Saccharomyces cerevisiae,* and interestingly, this complex is composed of four subunits, termed Hap2, Hap3, Hap4, and Hap5 (*Becker et al., 1991*). Among them, the Hap2 (NF-YA-like), Hap3 (NF-YB-like), and Hap5 (NF-YC-like) subunits form a heterotrimeric complex that is competent for DNA binding, whereas the additional Hap4 subunit is an acidic protein and harbors the transcriptional activation domain necessary for transcriptional

stimulation after interacting with the Hap2/Hap3/Hap5 complex (*McNabb and Pinto, 2005*). Interestingly, Hap4 is only present in fungi, and functional analyses in a variety of fungal species identified that homologs of Hap4, such as the *Aspergillus nidulans* HapX, the *Schizosaccharomyces pombe* Php4, the *Cryptococcus neoformans* homolog HapX, and *C. albicans* Hap43 were found to play both positive or negative roles in regulating the transcriptional responses to iron deprivation (*Jung et al., 2010*; *Singh et al., 2011*; *Skrahina et al., 2017*). Recently, there have been some progresses about the impact of Hap43 on the pathobiology of *C. albicans*. For example, Hap43 acts as a transcriptional repressor that is induced under low-iron conditions and required for iron-responsive transcriptional regulation and virulence since knocking out *HAP43* in *C. albicans* significantly upregulates the expression of genes involved in iron utilization under iron-limited conditions and attenuates virulence in a mouse model of disseminated candidiasis (*Chen et al., 2011*; *Hsu et al., 2011*). More importantly, genome-wide transcriptional profiling revealed that about 16% of the *C. albicans* ORFs were differentially regulated in a Hap43-dependent manner (*Chen et al., 2011*; *Singh et al., 2011*), and we found that a majority of differentially expressed genes (DEGs) are associated with oxidative stress and iron regulation, such as those involved in aerobic respiration, the respiratory electron transport chain, heme biosynthesis, and iron-sulfur cluster assembly, supporting the notion that Hap43 is one of the major iron-based redox sensors for *C. albicans* cells and contributes to the fine-tuned balance that adapts to different aspects of oxidative stress due to iron metabolism. Moreover, these data also raised a strong possibility that the regulatory function of Hap43 may be coupled to *C. albicans* commensalism by dealing with the cytotoxicity of ROS mainly generated in the iron-replete GI tract, in addition to its role in pathogenicity.

In this study, we sought to explore the underlying mechanism for a possible involvement of Hap43-dependent gene regulation in *C. albicans* gut commensalism, given that this commensal has to combat oxidative damage caused by excess iron content that is potentially detrimental for microbial cells. We unexpectedly unraveled an uncharacterized mechanism of post-translational modification of the iron-responsive repressor Hap43 that regulates adaptation of *C. albicans* to commensalism in the gut by ameliorating the iron-induced environmental oxidative stress.

## Results

### Deletion of *HAP43* significantly increases the commensal fitness of *C. albicans* in GI tract of mice fed a high-Fe diet

Accumulating evidence suggests the impact of the heterotrimeric CCAAT-binding complex on coordination of oxidative stress in fungi as the HAP complex in *S. cerevisiae* activates the expression of genes involved in oxidative phosphorylation in response to growth on non-fermentable carbon source (*Pinkham and Guarente, 1985*) and the homologous complex (AnCF) in *A. nidulans* is regulated at the post-translational level by the redox status of the cell and manipulates the expression of genes required for an appropriate response to oxidative stress (*Thön et al., 2010*). Moreover, microarray analyses in *C. albicans* showed that for 286 upregulated genes in *hap43Δ/Δ* relative to the wild type under iron-limiting conditions, 7.7 and 4.5 are those associated with aerobic respiration and the respiratory electron transport chain, respectively, highlighting the importance of Hap43 in iron-dependent oxidative stress (*Chen et al., 2011*). These observations prompt us to hypothesize that Hap43 may play an important role in regulating gastrointestinal commensalism of *C. albicans*, possibly by sensing changes of the oxidative status in this specific niche. To test this hypothesis, we first evaluated the contribution of Hap43 to the commensal fitness of *C. albicans* in GI tract using a well-established mouse model of stable GI candidiasis (*Chen et al., 2011*). Groups of female C57BL/6 mice receiving a normal Fe diet (NFD) (180 mg/kg Fe of diet) were inoculated by gavage with 1:1 mixtures of the wild type (WT) and *hap43Δ/Δ* mutant cells (*Figure 1A*). The relative abundance of each strain in fecal pellets was monitored by qPCR using strain-specific primers (*Supplementary file 1b*). Surprisingly, we did not observe significant differences in persistence between the WT and mutant (*Figure 1—figure supplement 1*). To investigate whether the inoculated *C. albicans* cells were really exposed to a host environment with high ROS levels, we examined ROS production in the colon tissue sections using the oxidant-sensitive fluorophore dihydroethidium (DHE). As shown in *Figure 1B and C*, the oxidative red fluorescence was almost undetectable in the colon (NFD), suggestive of insufficient ROS production. We therefore considered a possibility that the negligible effect on ROS generation could

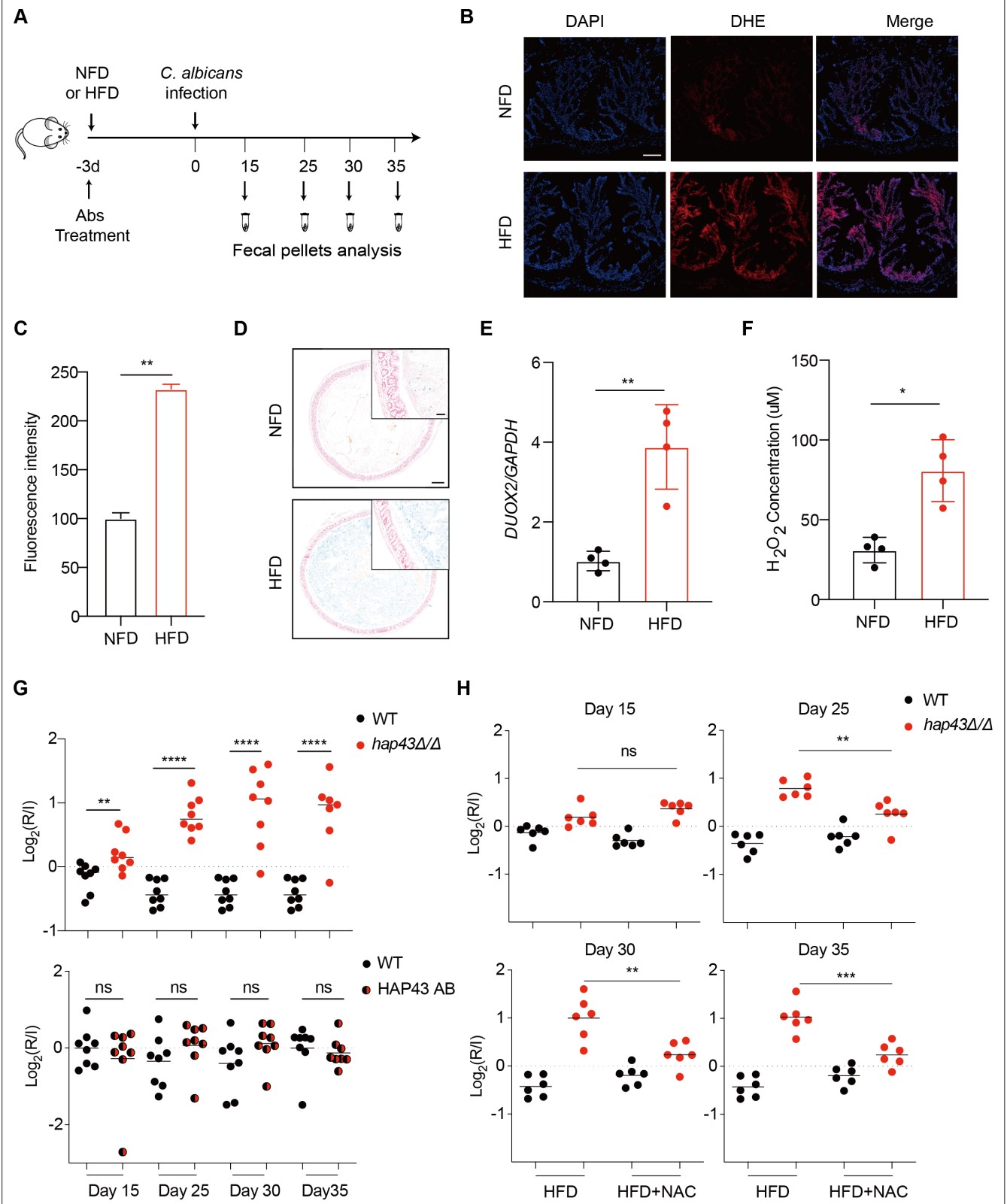

**Figure 1.** Deletion of *HAP43* significantly increases the commensal fitness of *C. albicans* in gastrointestinal (GI) tract of mice fed a high-Fe diet (HFD). (**A**) As depicted in the schematics, mice were fed a normal Fe (NFD) or HFD for 3 d prior to *C. albicans* inoculation. The mice continuously received the same diet during the course of experiments. (**B**) Colonic reactive oxygen species (ROS) accumulation in mice receiving an NFD or HFD diet for 3 d. Cryostat colonic sections were incubated with dihydroethidium (DHE) and DAPI. Scale bar, 100 μm. (**C**) Quantitative analysis using fluorescence intensity

*Figure 1 continued on next page*

*Figure 1 continued*

of DHE (**a**) in the colon. (**D**) Colonic samples were collected from mice fed either an NFD or HFD, formalin-fixed, paraffin-embedded, sectioned, and stained with Prussian blue for iron. Representative Prussian blue-stained colonic samples confirmed higher iron deposits in mice receiving HFD (iron blue, nucleus red). Scale bar, 200 μm; inset, 50 μm. (**E**) The expression of *DUOX2* mRNA in the colonic tissue of mice receiving NFD or HFD. Values were normalized to the expression levels of *GAPDH*. (**F**) The effect of iron on hydrogen peroxide ($H_2O_2$) levels in NFD or HFD-treated mice (n = 4 mice per group). (**G**) Mutant lacking *HAP43* exhibits enhanced commensal fitness in HFD-treated mice. Mice (n = 8 mice per group) fed an HFD were inoculated by gavage with 1:1 mixtures of the wild-type (WT) and either *hap43Δ/Δ* mutant or *HAP43* reintegrant (*HAP43* AB) cells (1 × 10⁸ CFU per mice). The fitness value for each strain was calculated as the $\log_2$ ratio of its relative abundance in the recovered pool from the host (R) to the initial inoculum (I), and was determined by qPCR using strain-specific primers that could distinguish one from another. (**H**) Treatment of the antioxidant N-acetyl-L-cysteine (NAC) is able to partially but significantly abrogate the commensal fitness of Hap43 mutant in HFD-fed mice. During the course of experiments, mice (n = 6 mice per group) fed HFD were received drinking water supplemented with or without NAC (1.5 g/L) and then inoculated by gavage with 1:1 mixtures of the wild-type (WT) and *hap43Δ/Δ* mutant cells (1 × 10⁸ CFU per mice). Results from three independent experiments are shown. All data shown are means ± SD. ns, no significance; *p<0.05, **p<0.01, ****p<0.0001; by unpaired Student's *t*-test (**C, E, F, G, H**).

The online version of this article includes the following figure supplement(s) for figure 1:

**Figure supplement 1.** Mutant lacking *HAP43* exhibits no change in commensal fitness in normal Fe diet (NFD)-treated mice.

**Figure supplement 2.** A high-Pi diet significantly promotes the intestinal colonization of *C. albicans*.

be attributed to inadequate iron bioavailability in the gut. To test the possibility, we modified our animal model by changing the mouse diet from the normal chow to a high-Fe diet (HFD) (400 mg/kg Fe of diet) (*Mahalhal et al., 2018*) since previous studies have shown that the amount of iron, which is about 2.2-fold higher than that in standard chow, is able to increase microbial exposure to iron without being overtly toxic to mice (*Mahalhal et al., 2018*; *Schwartz et al., 2019*). As expected, a 3-day HFD caused a significant increase of iron level in mouse colon, as determined by Prussian blue iron staining (*Figure 1D*). In line with the elevated level of iron, we clearly observed a marked increase of ROS levels in mouse colon after a an HFD as detected by DHE showing an increase in fluorescence (*Figure 1B and C*). The iron-induced ROS production in the gut was further confirmed by examining the transcript level of *DUOX2*. *DUOX2* encodes the dual oxidase 2, a hydrogen-peroxide generator at the apical membrane of gastrointestinal epithelia (*Donkó et al., 2014*). qRT-PCR results showed that mice fed the HFD had significant increase in *DUOX2* mRNA levels in the colon compared with mice on an NFD (*Figure 1E*). Finally, the hydrogen peroxide ($H_2O_2$) concentrations were determined in the mouse colon samples as an indicator of the level of oxidative stress (*Figure 1F*). We observed that treatment of mice with HFD significantly increased levels of $H_2O_2$, indicating the induction of oxidative stress pathways in the colon tissues. Taken together, these data strongly support that an HFD is sufficient to sustain a persistent exposure of gut microbes to high levels of ROS.

We first test whether HFD could affect gut colonization of *C. albicans* by comparing the fungal colonization levels in fecal samples from HFD or NFD-fed mice. Notably, a significant increase in fungal loads was observed in feces of the HFD-fed mice compared with those in the NFD group, at 5 d post-inoculation (*Figure 1—figure supplement 2A*). Consistently, this high enrichment trend of fungal cells in the gut of HFD-fed mice was further confirmed by both periodic acid-Schiff (PAS) and immunofluorescent staining assays (*Figure 1—figure supplement 2B*). Our data thus suggest that HFD rendered mice more susceptible to intestinal colonization of *C. albicans*. We next repeated the competitive gut infections in mice receiving HFD using the WT, *hap43Δ/Δ* mutant, and *HAP43* reintegrant (*HAP43* AB) strains (*Figure 1A*). Using this modified model, we found that a mutant lacking *HAP43* exhibited enhanced colonization fitness, such that the *hap43Δ/Δ* mutant cells significantly outcompeted WT *C. albicans* in 1:1 mixed infection (*Figure 1G*), implying a negative impact of the iron-responsive regulator Hap43 in gut commensalism of *C. albicans*. Interestingly, in vivo treatment of the antioxidant N-acetylcysteine (NAC) could reduce the competitive fitness of *hap43Δ/Δ* mutant in HFD-treated mice (*Figure 1H*). Taken together, our in vivo evidence highly suggests that Hap43 may play a negative role in regulating the gastrointestinal commensalism of *C. albicans*, especially under the circumstance in which the dietary stress is induced by ROS in the gut.

## High iron triggers Hap43 phosphorylation that is modulated by the protein kinase Ssn3

The commensal microbes colonized in the GI tract are thrived in comparatively high levels of iron because the majority of dietary iron is not absorbed, based on previous reports (*McCance and*

*Widdowson, 1938*; *Miret et al., 2003*). Deletion of the iron-responsive regulator Hap43 results in a beneficial effect on *C. albicans* colonization in the gut, making it highly possible that iron influences the expression of Hap43. Indeed, we found that by both qRT-PCR and immunoblotting, Hap43 in WT strain was significantly less expressed in iron-repleted (H) medium in comparison to the iron-depleted (L) medium (*Figure 2A and B*). Unexpectedly, immunoblot analysis of Hap43-Myc recovered from WT cells under iron replete vs. depleted condition identified an increase in the electrophoretic mobility of Hap43 in iron-replete medium compared to that under iron-depleted conditions (*Figure 2B*). Interestingly, when WT cells expressing Myc-tagged Hap43 were pre-grown to mid-exponential phase ($OD_{600}$ = 0.4–0.5) under iron-depleted conditions and then transferred into the iron-repleted medium (YPD), we found that a rapid gel mobility of Hap43 can be visualized at early time (2 min) after medium change (*Figure 2C*). We hypothesized that the shift in mobility on SDS-PAGE gel electrophoresis that is characteristic of the Hap43-Myc proteins might result from post-translational modification, for example, a covalent phosphorylation event. To test this possibility, WT cells expressing Myc-tagged Hap43 were grown to mid-exponential phase in either iron-rich medium or iron-depleted medium, and cell lysates were treated with or without lambda phosphatase, a broad specificity enzyme that acts on phosphorylated serine, threonine, and tyrosine residues. Immunoblotting analysis indicated that the mobility shifted form of Hap43-Myc reverted to the unshifted form if cell lysates were treated with lambda phosphatase, showing that the increased mobility induced by high iron was due to phosphorylation (*Figure 2D*).

We previously reported that the $Cys_6Zn_2$ DNA binding protein Sef1, another key player operating in the iron-regulatory circuit of *C. albicans*, was phosphorylated under iron-depleted conditions and the phosphorylation was catalyzed by the protein kinase Ssn3 (*Chen et al., 2012*). To test whether Hap43 phosphorylation under iron-replete conditions also could be modulated by the kinase activity of Ssn3, we expressed the Myc epitope-tagged version of Hap43 in *ssn3Δ/Δ* mutant strain and examined the mobility of Hap43 by immunoblotting. Compared to that of WT, the higher mobility form of Hap43 under iron-replete conditions was abolished in the mutant lacking *SSN3*, supporting the role of Ssn3 in phosphorylation of Hap43 (*Figure 2E*). An identical result was obtained when the mobility of Hap43-Myc was examined in the strain expressing a predicted kinase-dead allele of Ssn3 (Ssn3$^{D325A}$) (*Figure 2E*). Moreover, the putative enzyme–substrate interactions between Ssn3 and Hap43 was further reinforced through a co-immunoprecipitation assay, showing that Hap43-Myc was efficiently co-immunoprecipitated with Ssn3-TAP using either iron-replete or iron-depleted cells (*Figure 2F*).

## Ssn3-modulated phosphorylation induces cytoplasmic localization and protein degradation of Hap43 by ubiquitin-proteasome pathway

Studies have shown that Ssn3 acts as a cyclin-dependent protein kinase and catalyzes the phosphorylation of a number of specific transcription factors that strongly contributes to their transcriptional activities, nuclear-cytoplasmic localization, and/or stability (*Chi et al., 2001*; *Nelson et al., 2003*). The effect of Ssn3-modulated phosphorylation on subcellular localization of Hap43 was investigated by indirect immunofluorescence. Under iron-depleted conditions, the localization of Hap43-Myc was primarily nuclear in both WT and *ssn3Δ/Δ* mutant strains (*Figure 3A*). However, differences were observed under iron-replete conditions, in which the Hap43-Myc was found to be partially mislocalized from cytoplasm to nucleus in *ssn3Δ/Δ* mutant, compared to a complete cytoplasmic localization of this fusion protein in WT (*Figure 3A*). The intracellular localization of Hap43-Myc in either WT or *ssn3Δ/Δ* mutant strains was further analyzed by immunoblot analysis of cell fractions. Yeast nuclei were purified using a modified method described previously (*von Hagen and Michelsen, 2013*), and the analysis of Hap43-Myc distribution showed that Hap43 is only detected in the nuclear fraction of *ssn3Δ/Δ* mutant cells but not WT, when cultures were grown under iron-replete conditions (*Figure 3B*). These data highly suggested that Hap43 transcription factor is able to respond to iron status in *C. albicans* and modulates its expression and subcellular localization that is dependent on *SSN3* expression.

We note that loss of *SSN3* has a direct effect on the protein level of Hap43 when the cells were cultured in iron-replete conditions. Following the abolishment of increased mobility, the level of Hap43 is comparable with that found under iron-depleted conditions (*Figure 2E*). Importantly, the increased steady level of Hap43 protein could not be explained by its transcriptional level, as we observed that deletion of *SSN3* has no effect on the mRNA level of *HAP43* under iron-replete conditions (*Figure 3C*), suggesting that the post-translational modification by covalent phosphorylation may

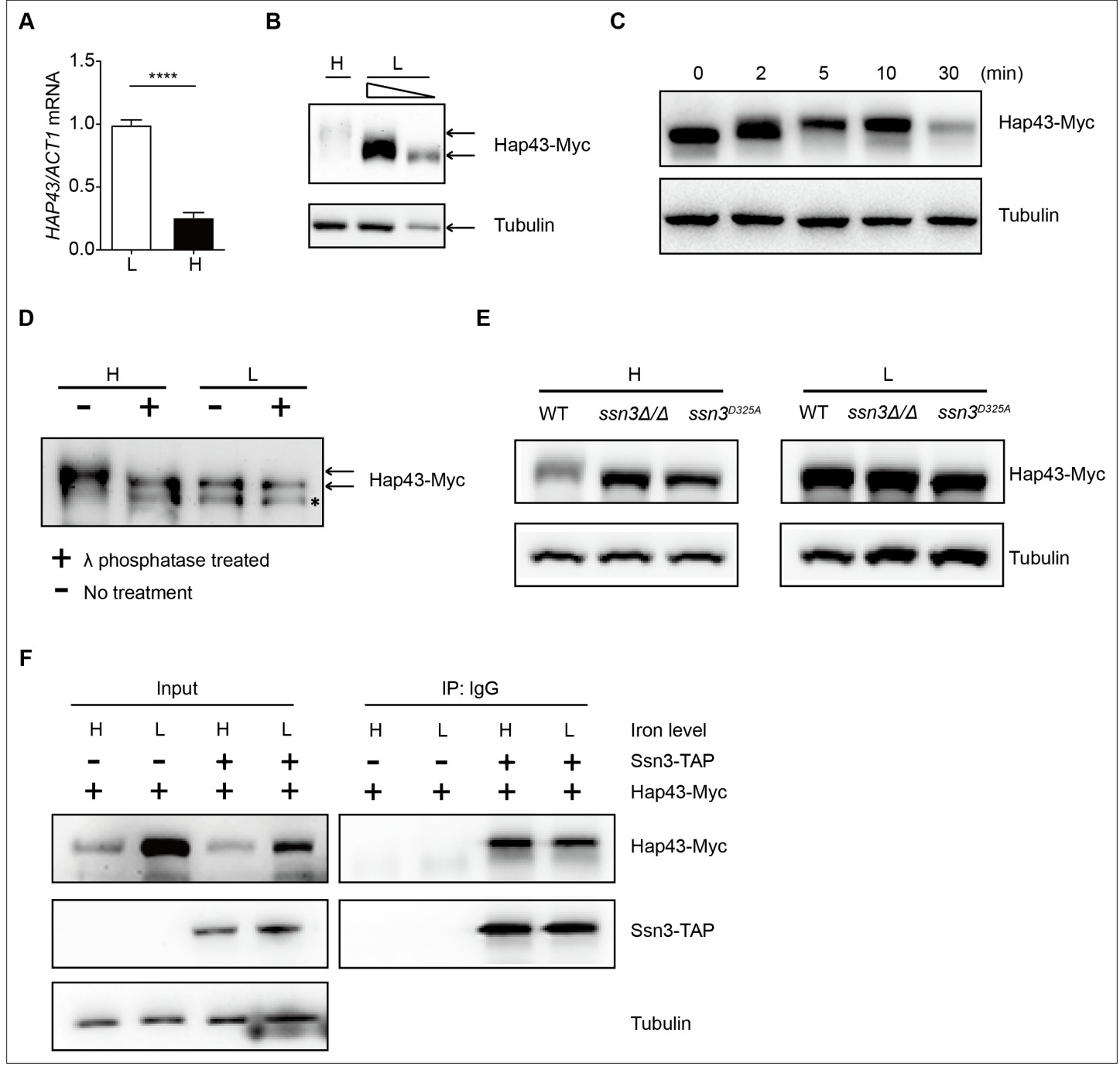

**Figure 2.** High iron triggers Hap43 phosphorylation that is modulated by the protein kinase Ssn3. (**A**) qRT-PCR analysis for *HAP43* mRNA in WT strain grown under iron-replete (H, high iron) or iron-depleted (L, low iron) conditions. Transcript levels were normalized to the level of *ACT1* mRNA. Results from three independent experiments are shown. All data shown are means ± SD. ****p<0.0001; by unpaired Student's *t*-test. (**B**) Immunoblots of C-terminally tagged Hap43 (Hap43-Myc) in WT cells propagated under iron-replete (H) or iron-depleted (L) conditions. To better display the mobility-shift on protein, we added additional lane and loaded smaller quantities of total proteins from low-iron culture. α-tubulin, internal standard. (**C**) Time course for electrophoretic mobility of Hap43-Myc in WT cells during a shift from iron-depleted to iron-replete conditions. (**D**) Immunoblots of purified Hap43-Myc protein either treated (+) or not treated (-) with λ phosphatase. Note that higher amounts of total proteins from high-iron cultures were loaded. * indicates a presumed Hap43-Myc C-terminal proteolysis product. (**E**) Immunoblots of Hap43-Myc recovered from WT, *ssn3Δ/Δ* or *SSN3^D325A^* cells under iron-replete (H) or iron-depleted (L) conditions. α-tubulin, internal standard. (**F**) Hap43-Myc is co-immunoprecipitated with Ssn3-TAP. WT strains containing only Ssn3-TAP or both Ssn3-TAP and Hap43-Myc were grown under iron-replete (H) or iron-depleted (L) conditions. Lysates were prepared under nondenaturing conditions, and IgG-sepharose affinity column was used to immune-precipitate Ssn3-TAP and interacting proteins.

The online version of this article includes the following source data for figure 2:

*Figure 2 continued on next page*

*Figure 2 continued*

**Source data 1.** Uncropped images of gels and blots in *Figure 2*.

promote protein instability of Hap43. To test this possibility, we employed the strains in which the sole Hap43-Myc allele is expressed under the control of the doxycycline (DOX)-inducible promoter (TetO-Hap43-Myc/*hap43Δ*) in either WT or *ssn3Δ/Δ* mutant backgrounds. Exponential-phase cells growing in iron-replete (YPD) medium supplemented with 50 μg/ml DOX were harvested, washed, and resuspended in fresh YPD medium, and whole-cell protein extracts were prepared at each time point for analysis by western blotting. Clearly, Hap43-Myc levels in WT were reduced by approximately 50% after 30 min incubation and continued to decline over the course of incubation (*Figure 3D*, left panel). In comparison, abundance of Hap43-Myc in *ssn3Δ/Δ* mutant remained at a relatively high level during the treatment (*Figure 3D*, right panel). These data highly suggested that Ssn3 significantly contributes to the protein stability of Hap43 under high iron conditions.

In eukaryotic cells, lysosomal proteolysis and the ubiquitin-proteasome system represent two major protein degradation pathways mediating protein degradation (*Lecker et al., 2006*). To clarify the exact proteolytic pathway implicated in Hap43 turnover under iron-replete conditions, we incubated cells with specific and selective inhibitors of the lysosome (Chloroquine) or the proteasome (MG132). Previous studies have shown that proteasome inhibitors such as MG132 are unable to penetrate WT yeast cells due to the impermeability of the cell wall or membrane, and therefore, mutant strains (e.g. *erg6Δ* and *pdr5Δ*) are required for experiments using the proteasome inhibitors since the mutant cells show increased drug permeability or reduced drug efflux (*Tumusiime et al., 2011*). We adapted the same strategy for inhibiting the proteasome and lysosome in *C. albicans*. A copy of *ERG6* gene was deleted from the strain in which the sole Myc-tagged version of Hap43 was expressed under the control of the DOX-inducible promoter (TetO-Hap43-Myc/*hap43Δ*), and the resulting mutant strain was treated with or without MG132. As shown in *Figure 3E*, treatment of mutant cells with 100 μM of MG132 for 30, 60, and 120 min significantly increased Hap43 protein levels compared with the untreated control. However, under the same experimental conditions, treating the cells with the lysosome inhibitor chloroquine had no effect on the decreased level of Hap43-Myc (*Figure 3—figure supplement 1*). Taken together, these data demonstrate that when *C. albicans* cells are grown under high iron conditions, the phosphorylated form of Hap43 is prone to be degraded through the proteasomal pathway.

To further verify this, we test a possibility of ubiquitination because this modification represents a common signal for proteasome-mediated protein degradation (*Hershko and Ciechanover, 1998*). In both WT and *ssn3Δ/Δ* mutant strain backgrounds (a copy of *HAP43* was C-terminally tagged with Myc epitope), we created strains that an epitope-tagged 3xHA-ubiquitin under the control of the DOX-inducible promoter was co-expressed with the Myc-tagged version of Hap43 (*Figure 3—figure supplement 2*). After a 6 hr induction using DOX (50 μg/ml), log-phase cells were collected and lysed, followed by immunoprecipitation of whole-cell extracts with anti-HA antibodies. Immunoblotting the precipitates with anti-Myc antibody revealed, as expected, a predominant band in WT but not *ssn3Δ/Δ* mutant (*Figure 3F*), indicating that only the phosphorylated form of Hap43-Myc was able to bind ubiquitin. In the other direction, Hap43 was fused C-terminally to a tandem affinity purification (TAP) tag in the WT strain (Hap43-TAP/Hap43) and lysates were immunoprecipitated with IgG beads to recover the TAP-tagged Hap43 and the precipitates were immunoblotted with K48 linkage-specific polyubiquitin antibodies, considering the fact that the polyubiquitin chains linked through K48 are the principal signal for targeting substrates to the proteasome for degradation (*Thrower et al., 2000*). A reactive smear, characteristic of polyubiquitination, was associated with immunoprecipitated TAP-tagged Hap43 (*Figure 3G*). Collectively, our data suggest that the Ssn3-modulated phosphorylation promotes protein degradation of Hap43 through a ubiquitin-proteasome pathway. It has to be mentioned that recent studies in *A. fumigatus* obtained similar observations. Similar to Hap43, the *A. fumigatus* homologue HapX also undergoes post-translational modifications (ubiquitination, sumoylation, and phosphorylation) during iron-replete conditions and these post-translational regulations are important to control iron homeostasis in *A. fumigatus* (*López-Berges et al., 2021*). Moreover, *A. fumigatus* HapX exhibits inverse regulatory activities by mediating repression or activation of vacuolar iron storage depending on the ambient iron availability, suggesting that HapX appears to be important for regulating both iron resistance and adaptation to iron starvation (*Gsaller et al., 2014*). Interestingly,

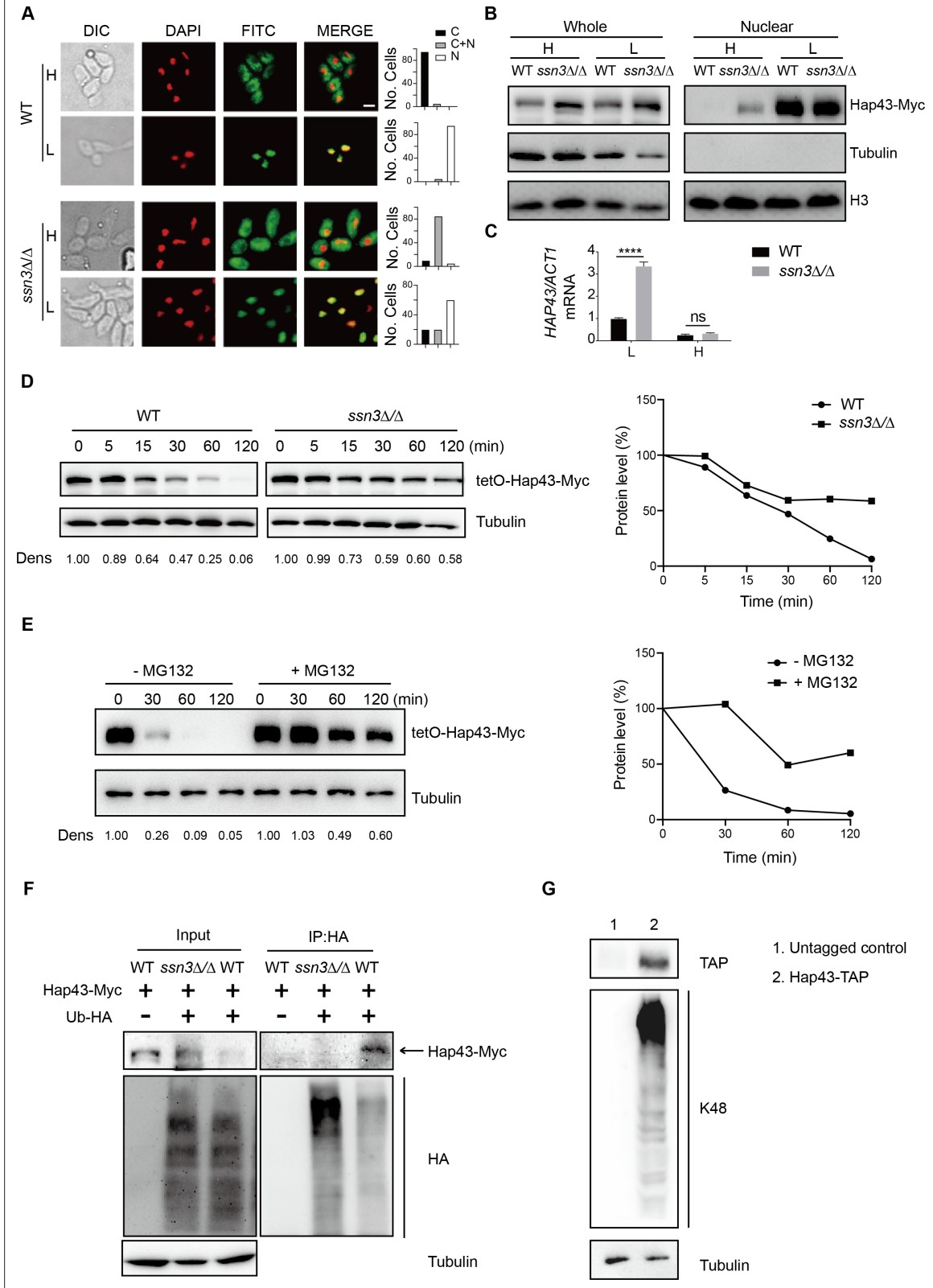

**Figure 3.** Ssn3-modulated phosphorylation induces cytoplasmic localization and protein degradation of Hap43 by ubiquitin-proteasome pathway. (**A**) Left panels: indirect immunofluorescence of Hap43-Myc in WT and *ssn3Δ/Δ* mutant strains grown under iron-replete (H, high iron) or iron-depleted (L, low iron) conditions. DIC represents phase images, DAPI represents nuclear staining, FITC represents Hap43-Myc staining, and Merge represents the overlay of Hap43-Myc and nuclear staining. Right panels: quantification of the cellular distribution of Hap43. Each bar represents the analysis of at least

*Figure 3 continued on next page*

*Figure 3 continued*

100 cells. C representing >90% cytoplasmic staining, N > 90% nuclear staining, and C + N a mixture of cytoplasmic and nuclear staining. Scale bar, 5 µm. (**B**) Immunoblots of Hap43-Myc in whole cell extracts and nuclear fraction of WT or *ssn3Δ/Δ* mutant cells propagated under iron-replete (H) or iron-depleted (L) conditions. Cellular contents were separated into cytosolic and nuclear fractions according to the protocol described in 'Materials and methods.' The nuclear marker H3 and cytoplasmic marker α-tubulin were used to display the purities of nucleus and cytoplasm. (**C**) qRT-PCR analysis for *HAP43* mRNA in WT and *ssn3Δ/Δ* strains grown under iron-replete (H) or iron-depleted (L) conditions. Transcript levels were normalized to the level of *ACT1* mRNA. Results from three independent experiments are shown. All data shown are means ± SD. ns, no significance; ****p<0.0001; by two-way ANOVA with Sidak's test. (**D**) Hap43 protein is stabilized in a *ssn3Δ/Δ* mutant. WT or *ssn3Δ/Δ* strains stably expressing doxycycline-inducible Myc-tagged Hap43 (TetO-Hap43-Myc) were treated with doxycycline. Cells were harvested in the exponential phase of growth, washed to remove doxycycline, and resuspended in fresh iron-replete medium. The turnover of Hap43-Myc in WT or *ssn3Δ/Δ* cells was then evaluated following the tetO promoter shut-off by removal of doxycycline through time-course experiments. Right panel: Hap43-Myc quantification after intensity analysis using ImageJ. (**E**) Similar to (**D**), after treatment with doxycycline, WT cells (a copy of *ERG6* was deleted) stably expressing doxycycline-inducible Myc-tagged Hap43 (TetO-Hap43-Myc) were harvested, washed, and treated with or without the proteasomal inhibitor MG132 (100 µM). The turnover of Hap43-Myc in WT cells was evaluated through time-course experiments. Right panel: Hap43-Myc quantification after intensity analysis using ImageJ. (**F**) Detection of Hap43 ubiquitination in *C. albicans*. WT and *ssn3Δ/Δ* mutant strains were engineered by stably expressing either Hap43-Myc alone or both Hap43-Myc plus tetO-HA-Ub. Both strains were incubated under iron-replete plus 50 µg/ml doxycycline conditions and cell extracts were subjected to immunoprecipitation with anti-HA-conjugated beads followed by western blot analysis with anti-Myc antibodies for detection of ubiquitinated Hap43. (**G**) Detection of Hap43 polyubiquitination in *C. albicans*. The WT strain was engineered by stably expressing Hap43-TAP and grown under iron-replete conditions. Cell extracts were immunoprecipitated with IgG-sepharose followed by western blot analysis with anti-K48 linkage antibody for detection of K48-linked polyubiquitination of Hap43.

The online version of this article includes the following source data and figure supplement(s) for figure 3:

**Source data 1.** Uncropped images of gels and blots in *Figure 3*.

**Figure supplement 1.** Chloroquine had no effect on Hap43 degradation under high iron conditions.

**Figure supplement 1—source data 1.** Uncropped images of gels and blots in *Figure 3—figure supplement 1*.

**Figure supplement 2.** Immunoblots showing the induction of an epitope-tagged 3xHA-ubiquitin under the control of the doxycycline (DOX)-inducible promoter.

**Figure supplement 2—source data 1.** Uncropped images of gels and blots in *Figure 3—figure supplement 2*.

*C. albicans* Hap43, in stark contrast to HapX of *A. fumigatus*, appears to play only a minor role in mediating the adaption to iron excess, although this factor did act positively for the activation of iron uptake genes during iron starvation (*Skrahina et al., 2017*).

## Identification of the Hap43 phosphorylation sites that signal its ubiquitination and degradation

Together with the aforementioned results that the iron-responsive regulator Hap43 is phosphorylated in *C. albicans* cells grown under iron-replete conditions, this observation prompts us to identify at which serine/threonine residues Hap43 is phosphorylated. First, we started with an in silico prediction by using the Kinasephos 2.0 server (http://kinasephos2.mbc.nctu.tw/; *Wong et al., 2007*), and this analysis predicted 12 putative serine/threonine phosphorylation sites within Hap43 of *C. albicans*. Moreover, Ssn3 of *C. albicans* is orthologous to *S. cerevisiae* Srb10, a cyclin-dependent kinase subunit of the Cdk8 module of Mediator (*Björklund and Gustafsson, 2005*), and putative Cdk8-dependent phosphorylation sites identified to date are serine/threonine residues flanked by a proline 1–2 residues toward the C-terminus and/or by a proline 2–4 residues toward the N terminus (*Chi et al., 2001*). We examined the Hap43 sequence and identified another 17 potential Ssn3 kinase phosphorylation sites that meet the criteria described above (*Figure 4A*). To experimentally confirm these in silico predictions, we generated amino acid substitution mutants in which the neutral amino acid alanine replaced serine/threonine at the predicted 12 putative phosphorylation sites to change the conserved phosphorylation motif in order to mimic the dephosphorylated state of Hap43. By use of the strain (TetO-Hap43-Myc/*hap43Δ*) described in *Figure 3D*, where the sole Hap43-Myc allele driven by the DOX-inducible promoter was expressed in the *hap43Δ/Δ* strain background, we successfully created seven mutants, each of which included one or two mutated S/T (to Ala) sites. Similarly, exponential-phase cells growing in iron-replete (YPD) medium supplemented with 50 µg/ml DOX were harvested, washed, and resuspended in fresh YPD medium, and whole-cell protein extracts were prepared at indicated time points or after 2 hr of incubation, and analyzed by western blotting for the phosphorylation status and overall level of Hap43. Intriguingly, we found that single or double mutation of the

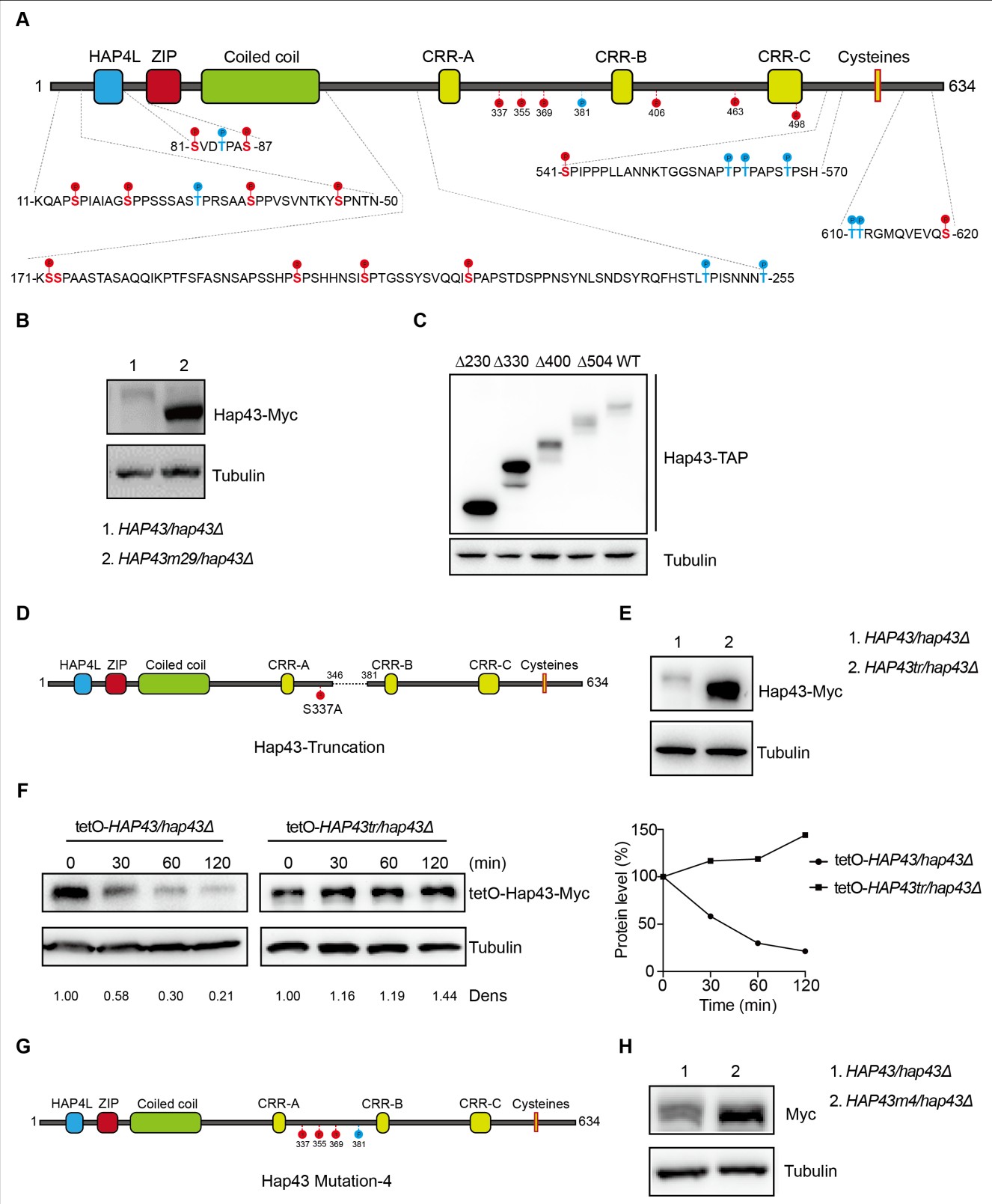

**Figure 4.** The critical phosphorylation sites are essential for Hap43 stabilization. (**A**) Schematic representation of *C. albicans* Hap43. Putative phosphorylation sites predicted by the Kinasephos 2.0 server and Cdk8-dependent phosphorylation sites are represented. (**B**) Immunoblots of Hap43-Myc in strains expressing either the WT or the amino acid mutation (*HAP43m29*; all 29 putative S/T phosphorylation sites were replaced with alanine residues) allele of Hap43. Cells were treated at high iron conditions. (**C**) Immunoblots of Hap43-TAP in WT and truncation mutant strains grown under

*Figure 4 continued on next page*

*Figure 4 continued*

iron-replete conditions. (**D**) Schematic representation of *C. albicans* Hap43 truncation. Hap43 truncation mutation (*HAP43tr*) was generated by deleting the 36 residues (346–381 aa) of Hap43 in HAP43$^{S337A}$ strain. (**E**) Immunoblots of Hap43-Myc in strains expressing either the WT or the truncation mutation (*HAP43tr*) allele of Hap43. Cells were treated at high iron conditions. (**F**) Strains expressing either the WT or the truncation mutation (*HAP43tr*) allele of Hap43 under control of the inducible tetO promoter were treated with doxycycline. Cells were harvested in the exponential phase of growth, washed to remove doxycycline, and resuspended in fresh iron-replete medium (YPD). The turnover of Hap43-Myc in WT or truncation mutant cells was then evaluated following the tetO promoter shut-off by removal of doxycycline, through time-course experiments. Right panel: Hap43-Myc quantification after intensity analysis using ImageJ. (**G**) Schematic representation of *C. albicans* Hap43 mutation-4 (*HAP43*m4). Four putative S/T phosphorylation sites (S337/S355/S369/T381) in Hap43 were individually replaced with alanine residues. (**H**) Immunoblots of Hap43-Myc in strains expressing either the WT or the amino acid substitution (*HAP43*m4) allele of Hap43. Similar to (**E**), cells were treated at high iron conditions.

The online version of this article includes the following source data and figure supplement(s) for figure 4:

**Source data 1.** Uncropped images of gels and blots in *Figure 4*.

**Figure supplement 1.** The Hap43 mutants harboring serine/threonine-to-alanine substitutions in its one or two putative phosphorylation sites showed the WT-like degradation patterns of Hap43 under high iron conditions.

**Figure supplement 1—source data 1.** Uncropped images of gels and blots in *Figure 4—figure supplement 1*.

**Figure supplement 2.** The mutants harboring amino acid substitutions or fragment truncation showed no defects in vegetative growth.

**Figure supplement 3.** The critical phosphorylation region is essential for Hap43 stabilization.

**Figure supplement 4.** The mutants harboring amino acid substitutions or fragment truncation showed no defects in vegetative growth.

**Figure supplement 5.** The *HAP43* mutant-4 strain (*HAP43m4/hap43Δ*) harboring amino acid substitutions showed no defects in vegetative growth.

---

predicted phosphorylation sites had no change of the phosphorylation pattern and consequently still promoted protein degradation of Hap43 as the WT cells did (***Figure 4—figure supplement 1A and B***), indicating that phosphorylation of Hap43 should not occur in merely one or two residues.

We therefore generated a *HAP43* mutant strain (Hap43m-Myc/*hap43Δ*) that all 29 putative S/T phosphorylation sites, including the 12 residues predicted by computer algorithms and 17 residues matching the Cdk8 consensus phosphorylation sites, were replaced with alanine residues (***Figure 4A***). An immunoblot with cell lysates from both WT (Hap43-Myc/*hap43Δ*) and *HAP43* mutant-29 (Hap43m29-Myc/*hap43Δ*) clearly revealed that the replacement of all 29S/T residues abolished the upward shift (phosphorylation) of Hap43-Myc band induced by high iron and as a result significantly increased the steady level of Hap43 (***Figure 4B***). As a control, amino acid replacement appeared to have no effect on the growth and function of *HAP43* mutant harboring 29-point mutations under low iron conditions (***Figure 4—figure supplement 2***). Taken together, our experiments identified multiple S/T residues as important Hap43 phosphorylation sites in vivo.

Another alternative strategy for the role of phosphorylation is to assess the degradation of truncated Hap43. Four kinds of C-terminally deleted *HAP43* ORFs fused with the TAP tag were generated and introduced into *hap43Δ/Δ* mutant (***Figure 4—figure supplement 3A***). We showed that there was no significant difference in the transcript levels of WT and truncated *HAP43* (***Figure 4—figure supplement 3B***); however, their protein levels varied dramatically (***Figure 4C***). Among them, Hap43 truncating mutations (Δ400 and Δ504) give rise to almost similar levels as the full length of WT, whereas the mutation (Δ330) results in a suddenly dramatic increase of Hap43 level. These results strongly suggest that the region within residues 330–400 harbors the signal contributing the phosphorylation-dependent degradation of Hap43. To further verify this, we deleted the 36 residues (346–381 aa) of Hap43 in HAP43$^{S337A}$ strain (make sure there is no T or S left between 300-400aa) and generated a Hap43 truncation mutant (TetO-Hap43tr-Myc/*hap43Δ*) (***Figure 4D***). Similar to the phenotypes observed in the *HAP43* mutant, deletion of the 36 amino acid residues also leads to increased level of the truncated form of Hap43 (***Figure 4E***) and abrogated the high iron-induced protein degradation (***Figure 4F***). As a control, fragment deletion appeared to have no effect on the growth and function of *HAP43* truncation mutant under low iron conditions (***Figure 4—figure supplement 4***).

By combining the results shown above, we finally focused on the four putative phosphorylation sites (S337/S355/S369/T381) between residue 336 and 381. To verify this, we generated a *HAP43* mutant-4 strain (Hap43m4-Myc/*hap43Δ*) in which site-directed mutagenesis was used to convert the codons for serine and tyrosine at these sites to codons for alanine (***Figure 4G***). Consistently, we found that under high iron conditions, the mutant exhibited significantly higher level of Hap43 proteins than that of the wild type (***Figure 4H***). As a control, replacement of these four residues with alanine

appeared to have no effect on the growth and function of Hap43 (*Figure 4—figure supplement 5*). Collectively, our data identified potential phosphorylation sites responsible for protein instability of Hap43 when *C. albicans* cells were grown under high iron conditions.

## Importance of Hap43 phosphorylation for alleviating Fenton reaction-induced ROS toxicity

Numerous studies have demonstrated that bivalent iron cation drives the Fenton reaction ($Fe^{2+}$ + $H_2O_2$ $Fe^{3+}$ +.OH + $OH^-$) that plays an important role in the transformation of poorly reactive radicals into highly reactive ones, leading to many disturbances contributing to cellular toxicity (*Ryan and Aust, 1992*). The Hap complex, which is composed of Hap2, Hap3, Hap5, and Hap43 in *C. albicans*, has been found to play a key role in connecting the iron acquisition to oxidative stress response by regulating the expression of oxidative stress genes (e.g. *CAT1*, *SOD4*, *GRX5*, and *TRX1*), those who have been known to be induced in the production of ROS under iron-overloaded conditions (*Chakravarti et al., 2017*; *Mao and Chen, 2019*). We therefore hypothesized that Hap43 phosphorylation may play a role in the coordinate regulation of *C. albicans* against iron-induced ROS toxicity. To test this, we first measured the intracellular ROS production in *C. albicans* cells grown under YPD or YPD supplemented with 200 µM $FeCl_3$ conditions by a fluorometric assay using hydroxyphenyl fluorescein (HPF; 5 µM) (*Avci et al., 2016*). As shown in *Figure 5A and B*, ROS levels were moderately elevated in *C. albicans* cells after incubation in YPD medium, whereas massive increase was observed in medium supplemented with $FeCl_3$, to a level comparable to that observed in medium with $H_2O_2$. As controls, iron-induced ROS production via Fenton reaction could be prevented by treating the cells with the antioxidant N-acetyl-L-cysteine (NAC). These data clearly indicated that high levels of iron are sufficient to significantly enhanced ROS production in *C. albicans*. More importantly, we observed that iron-triggered degradation of Hap43 could be inhibited by treating the cells with NAC (*Figure 5C*) and treatment of *C. albicans* cells with menadione, an inducer of endogenous ROS, leads to the reduction of the Hap43 protein level (*Figure 5D*), supporting the proposition that the promotion of iron-induced generation of ROS may account for the *SSN3*-dependent ubiquitination and phosphorylation-dependent degradation of Hap43.

To further test the potential role of Hap43 phosphorylation in protecting *C. albicans* cells from ROS-induced cytotoxicity, we examined the growth of different strains (WT, *hap43Δ/Δ*, Hap43-Myc/*hap43Δ*, Hap43m29-Myc/*hap43Δ*, Hap43tr-Myc/*hap43Δ*, and Hap43m4-Myc/*hap43Δ* mutants) in medium supplemented with or without $H_2O_2$. Compared to the WT, deletion of *HAP43* showed remarkable resistance to $H_2O_2$ (*Figure 5—figure supplement 1*), suggesting that loss of Hap43 promotes cell survival under oxidative stress. Actually, the observation is consistent with our in vivo fitness study showing an increased competitive ability of the *hap43Δ/Δ* mutant to colonize the GI tract (*Figure 1*). Interestingly, we also found that compared to the WT (Hap43-Myc/*hap43Δ*), abolishment of Hap43 phosphorylation in each of the three mutants generated above, including the Hap43 truncation mutant (Hap43tr-Myc/*hap43Δ*), *HAP43* mutant-29 (Hap43m29-Myc/*hap43Δ*), or *HAP43* mutant-4 (Hap43m4-Myc/*hap43Δ*), showed significantly greater sensitivity to oxidative stress (*Figure 5E–G*), arguing that the ubiquitin-dependent degradation of Hap43 after phosphorylation contributes to the protection against ROS-induced cytotoxicity. This notion was further supported by the in vivo evidence that the *HAP43* mutant-29, Hap43 truncation, and *HAP43* mutant-4 mutants could be outcompeted by the WT strain when cells stably colonize in mouse GI tract (*Figure 5H–J*). Taken together, our data highly suggest that iron-induced Hap43 phosphorylation, followed by ubiquitin-dependent proteasomal degradation, acts to protect *C. albicans* from ROS toxicity and thus promote its survival in GI tract, a niche normally considered as an iron-replete environment.

## Iron-induced phosphorylation and degradation of Hap43 leads to de-repression of antioxidant genes

ROS generation in actively growing cells occurs via Fenton reaction or as a by-product of mitochondrial respiration. Previous studies have shown that Hap43 is primarily a transcriptional repressor and enriched in the nucleus in response to iron depletion, particularly responsible for repression of genes that encode iron-dependent proteins involved in mitochondrial respiration and iron-sulfur cluster assembly (*Chen et al., 2011*; *Hsu et al., 2011*). Moreover, our data revealed that iron-triggered post-translational modification of Hap43, including cytoplasmic localization, phosphorylation,

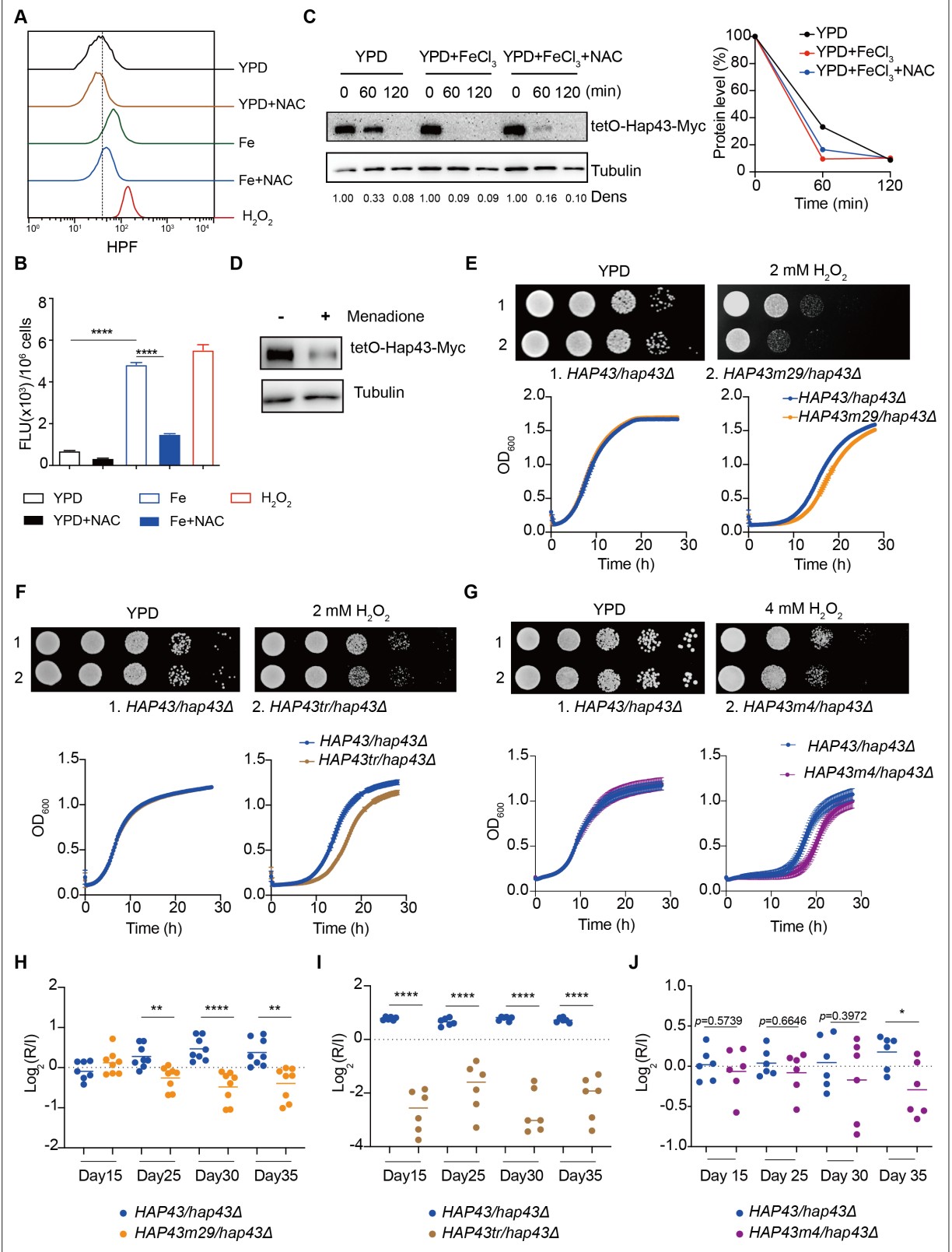

**Figure 5.** Hap43 phosphorylation is important for alleviating Fenton reaction-induced reactive oxygen species (ROS) toxicity and for gastrointestinal (GI) colonization. (**A, B**) Intracellular ROS production of *C. albicans* under different experimental conditions. *C. albicans* yeast cells were grown on YPD supplemented with indicated reagents. About $1 \times 10^7$ cells in exponential growth phase were collected, washed with PBS, stained with 5 mM of HPF, and analyzed using FACS (**A**) or the microplate reader (**B**). (**C, D**) Hap43 stability assay by immunoblots in WT strain stably expressing doxycycline-

*Figure 5 continued on next page*

*Figure 5 continued*

inducible Myc-tagged Hap43 (TetO-Hap43-Myc). Exponential-phase cells grown in iron-replete (YPD) medium supplemented with 50 μg/ml doxycycline were harvested, washed, and resuspended in fresh YPD medium only, YPD supplemented with 200 μM FeCl$_3$, a combination of 200 μM FeCl$_3$ and 20 mM N-acetyl-L-cysteine (NAC) (**C**) or 20 μM menadione for 120 min (**D**). (**E–G**) Growth of different *C. albicans* strains, including WT (*HAP43/hap43Δ*), *HAP43m29* mutant (**E**) or *HAP43tr* truncation mutant (**F**) or *HAP43m4* mutant (**G**), under oxidative stresses. Top panel: strains were spotted with tenfold serial dilutions onto YPD or YPD supplemented with 2 mM H$_2$O$_2$ and grown for 2 d at 30°C. Bottom panel: growth curve analysis of strains in YPD liquid medium supplemented with 2 mM or 4 mM H$_2$O$_2$ at 30°C. OD$_{600}$ readings were obtained every 15 min in a BioTek Synergy 2 Multi-mode Microplate Reader. (**H–J**) Each of the *HAP43* mutants, including *HAP43m29* mutant (**H**), *HAP43tr* truncation mutant (*Hap43tr*) (**I**), or *HAP43m4* mutant (**J**), exhibits decreased commensal fitness in mice. Similar to *Figure 1F*, mice (n = 6 or 8) were inoculated by gavage with 1:1 mixtures of the WT and each of the *Hap43* mutant strains (1 × 10$^8$ CFU per mice). The fitness value for each strain was calculated as the log$_2$ ratio of its relative abundance in the recovered pool from the host (**R**) to the initial inoculum (**I**), and was determined by qPCR using strain-specific primers that could distinguish one from another. Results from three independent experiments are shown. All data shown are means ± SD. *p<0.05; **p<0.01; ****p<0.0001; by one-way ANOVA with Sidak's test (**B**) or unpaired Student's *t*-test (**H–J**).

The online version of this article includes the following source data and figure supplement(s) for figure 5:

**Source data 1.** Uncropped images of gels and blots in *Figure 5*.

**Figure supplement 1.** Growth of the *hap43Δ/Δ* mutant under oxidative stresses.

ubiquitination, and proteasomal degradation, heavily impacts the ability of *C. albicans* to adapt and respond to oxidative stress. These findings are very informative and prompt us to examine whether the phosphorylation-dependent degradation of Hap43 may correlate with activation of antioxidant response. In other words, it is likely that iron-induced post-translational modification of Hap43 may directly cause ROS elimination by upregulating the expression of antioxidant genes when *C. albicans* cells are bathed under conditions of high iron availability.

Given that the iron-responsive transcription factor Hap43 undergoes ubiquitin-dependent proteasomal degradation after phosphorylation, we provided evidence that deletion of the protein kinase Ssn3 prevents its degradation and causes nuclear mislocalization, when *C. albicans* cells are grown under iron-replete conditions (*Figure 3A and B*). Consistently, replacement of either 29 or 4S/T residues by alanine, as well as the truncated form, were found to abrogate phosphorylation and degradation of Hap43 (*Figure 4*), prompting us to hypothesize that the unphosphorylated form of Hap43 through either amino acid substitutions or truncation may alter its cellular localization when cells are grown under iron-replete conditions. To test this hypothesis, indirect immunofluorescence of formaldehyde-fixed yeast cells from WT (Hap43-Myc/*hap43Δ*), *HAP43* mutant-29 (Hap43m29-Myc/*hap43Δ*), Hap43 truncation mutant (Hap43tr-Myc/*hap43Δ*), or *HAP43* mutant-4 (Hap43m4-Myc/*hap43Δ*) strain, at the early mid log phases of growth on YPD supplemented with FeCl$_3$, was used to examine the subcellular localization of Hap43. As shown in *Figure 6A*, *Figure 6—figure supplement 1A*, and *Figure 6—figure supplement 2A*, WT Hap43 localized to the cytoplasm, while unphosphorylated form of Hap43 (Hap43tr, Hap43m29, and Hap43m4) localized to the nucleus, suggesting that abolishing the phosphorylation-dependent modification resulted in relocation of Hap43 from cytoplasm to nucleus.

The antioxidant enzyme-mediated adaptive response has been demonstrated to attenuate toxicity caused by oxidative stress and a list of enzymes, including catalases, superoxide dismutases, peroxidases, and peroxiredoxins, have been found to be the most ubiquitous effectors in microbial eukaryotes (*Aguirre et al., 2005*). Moreover, sequence analysis demonstrated the presence of CCAAT *cis*-acting element, a conserved Hap43 DNA recognition motif, on the promoter regions of antioxidant genes *CAT1*, *SOD2*, *GSH1,* and *TRR1*. We therefore ask whether the unphosphorylated form of Hap43, once located to the nucleus, has the DNA binding capacity. ChIP-qPCR assays were performed to investigate Hap43 binding to the promoter sequences containing CCAAT motifs in the selected antioxidant genes. As expected, the mutated or truncated Hap43 (Hap43tr or Hap43m29) significantly enriched in the promoter regions of the four target antioxidant genes (*Figure 6B* and *Figure 6—figure supplement 1B*), which suggested a direct regulation of ROS detoxification in *C. albicans* by post-translational modification of Hap43. Indeed, when the expression levels of these four antioxidant genes were examined by qPCR, we found that upon binding to the promoters directly, Hap43 significantly repressed the expression of *CAT1* and *SOD2* in Hap43tr, Hap43m29, and Hap43m4 strains (*Figure 6C*, *Figure 6—figure supplement 1C*, and *Figure 6—figure supplement 2B*). In some cases, we did not observe significant changes in expression of several indicated antioxidant genes such as *TRR1* and *GSH1*, even though we are aware of a highly significant increase in Hap43 binding

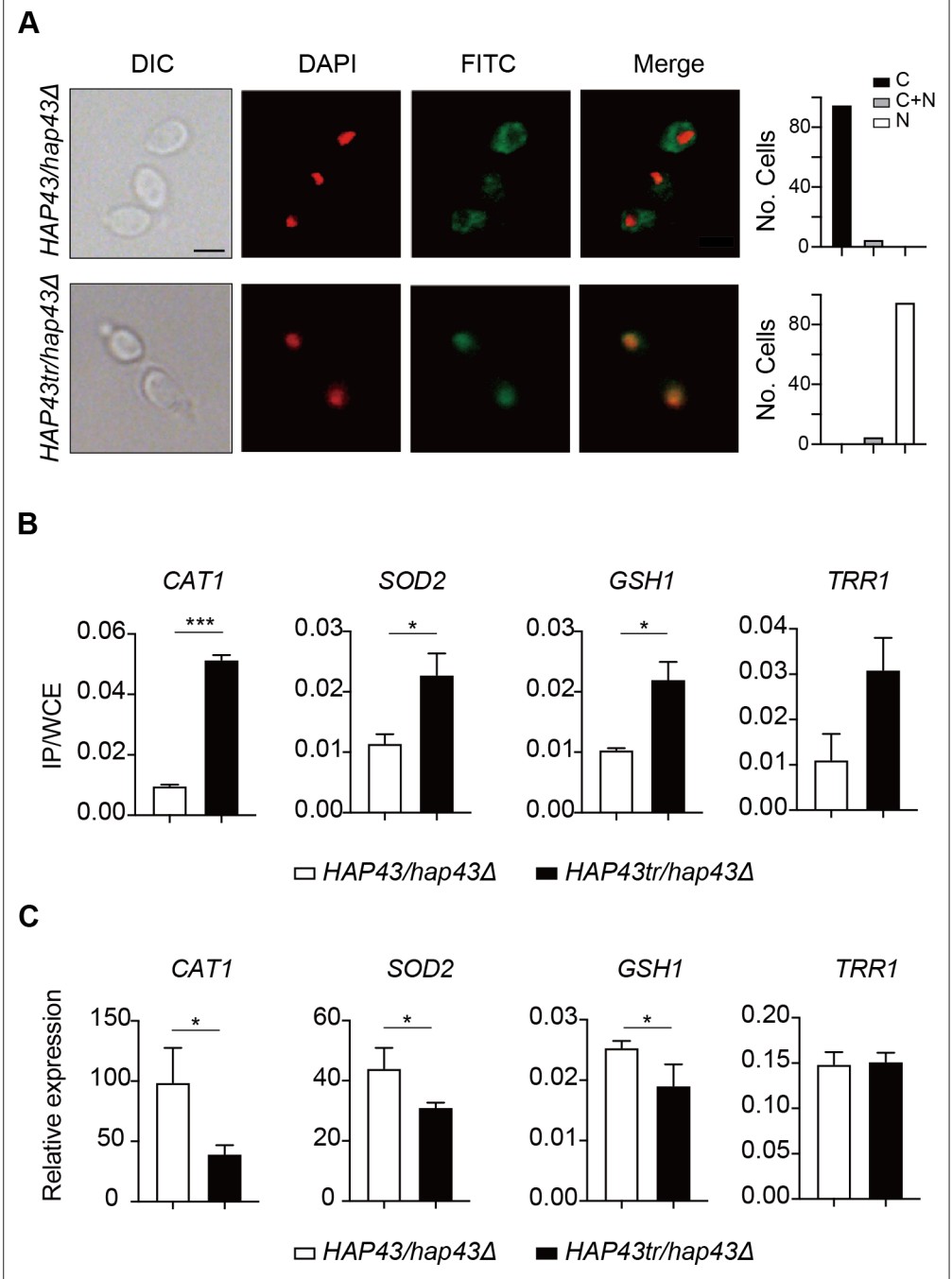

**Figure 6.** Iron-induced phosphorylation and degradation of Hap43 leads to de-repression of antioxidant genes. (**A**) Left panels: indirect immunofluorescence of Hap43-Myc in *HAP43/hap43Δ* and *HAP43tr/hap43Δ* strains grown under iron-replete conditions. DIC represents phase images, DAPI represents nuclear staining, FITC represents Hap43-Myc staining, and Merge represents the overlay of Hap43-Myc and nuclear staining. Right panels: quantification of the cellular distribution of Hap43. Each bar represents the analysis of at least 100 cells. C representing >90% cytoplasmic staining, N > 90% nuclear staining, and C + N a mixture of cytoplasmic and nuclear staining. Scale bar, 5 µm. (**B**) ChIP of Hap43-Myc on the promoters that contain CCAAT boxes in a set of antioxidant genes. Overnight cultures of WT (*HAP43/hap43Δ*) and truncation mutant (*HAP43tr/hap43Δ*) cells were diluted in YPD plus 400 mM FeCl₃ and grown to log phase at 30°C before formaldehyde. Enrichment is presented as a ratio of qPCR of the indicated gene promoter IP (bound/input) over an *ACT1* control region IP (bound/input) of the tagged strain, further normalized to the control strain. (**C**) qRT-PCR analysis for mRNA levels of a set of antioxidant genes in WT (*HAP43/hap43Δ*) and truncation mutant (*HAP43tr/hap43Δ*) strains grown under iron-

*Figure 6 continued on next page*

*Figure 6 continued*

replete conditions. Transcript levels were normalized to the level of *ACT1* mRNA. Results from three independent experiments are shown. All data shown are means ± SD. ns, no significance; *p<0.05; **p<0.01; ***p<0.001; by unpaired Student's *t*-test (**B, C**).

The online version of this article includes the following figure supplement(s) for figure 6:

**Figure supplement 1.** Iron-induced phosphorylation and degradation of Hap43 leads to de-repression of antioxidant genes.

**Figure supplement 2.** Iron-induced phosphorylation and degradation of Hap43 leads to de-repression of antioxidant genes.

to their promoters based on our ChIP assays, arguing the involvement of other regulators, such as Cap1 and Tsa1/Tsa1B (*Urban et al., 2005*; *Wang et al., 2006*), in modulating the transcript levels of these genes, in addition to Hap43.

Taken together, our data proposed a model (*Figure 7*) that the iron-induced post-translational modification of Hap43, including cytoplasmic localization, Ssn3-dependent phosphorylation, ubiquitination, and proteasomal degradation, results in the de-repression of antioxidant genes (e.g. *CAT1*

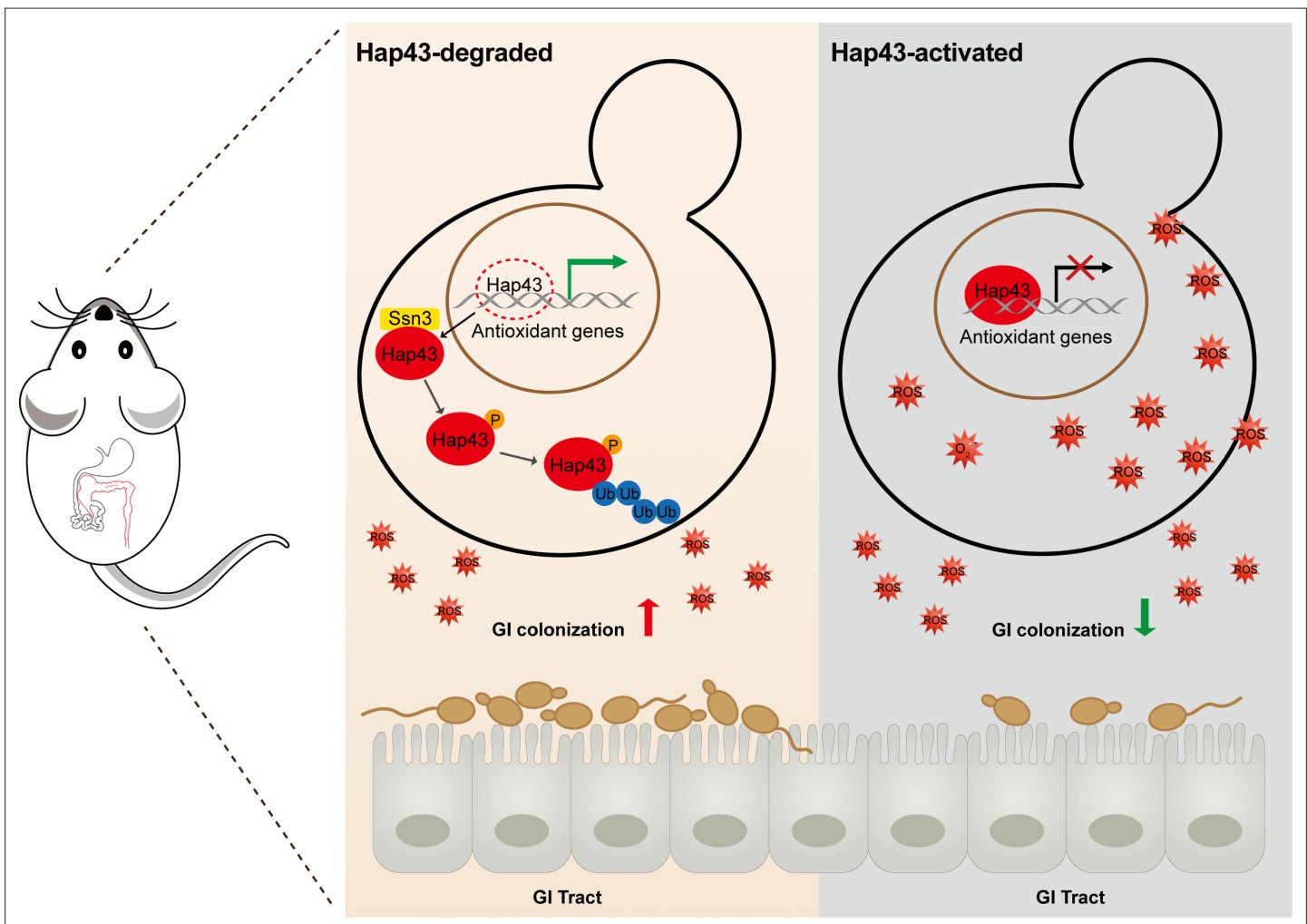

**Figure 7.** Model for the role of post-translational medication of Hap43 in promoting gastrointestinal (GI) commensalism of *C. albicans*. In the iron-rich environment such as GI tract, the iron-responsive regulator Hap43 is subject to covalent post-translational modifications, including phosphorylation and ubiquitination, and causes cytoplasm-nuclear relocation and protein degradation via proteasome activity, thus serving as a positive signal to de-repress the expression of a set of antioxidant genes (e.g. *CAT1* and *SOD2*), an event that is most effective in lowering cytotoxicity induced by iron-mediated reactive oxygen species (ROS) production and promotes *C. albicans* commensalism in GI tract.

and *SOD2*), an event that is most effective in lowering cytotoxicity induced by oxidative stress and promotes *C. albicans* commensalism in GI tract.

## Discussion

Iron makes an ideal redox active cofactor for a variety of key biological processes and therefore becomes an indispensable element for all eukaryotes and the vast majority of prokaryotes. However, studies have revealed that iron excess is able to promote the production of potentially harmful ROS through accelerating the Fenton reactions, causing deleterious cellular effects such as lipid peroxidation, protein oxidation and carbonylation, and DNA mutagenesis and destabilization (*Galaris et al., 2019*). The need to avoid oxidative damages is particularly acute in the case of human fungal pathogens like *C. albicans*, mainly because these microbes are often subjected to assault by ROS produced by iron metabolism, environmental competitors, or phagocytic cells during infections, as well as the endogenously produced ROS. Here, we discovered an uncharacterized detoxification strategy that *C. albicans* used to combat the toxic effects of ROS accumulation and promote its colonization in GI tract. Our data highly suggest that the iron-dependent global regulator Hap43, through a previously unknown post-translational modification mechanism, senses the iron status of the cell and negatively regulates the gene expression of antioxidant enzymes.

To combat the iron sequestration by host iron-binding proteins, commensal microbes such as *C. albicans* have concomitantly evolved elaborate mechanisms to acquire iron, which play an important role in enabling the fungus to survive phagocytosis and proliferate inside macrophages by utilizing their intracellular storages of iron (*Nevitt and Thiele, 2011*; *Seider et al., 2014*). These iron-scavenging strategies could be summarized as the following categories: (1) the reductive iron uptake pathway requiring the cell-surface ferric reductases such as Fre7, Fre10, and Cfl1, and a high-affinity heterodimeric transport complex that consists of the multicopper ferroxidase Fet34 and the iron permease Ftr1; (2) the heme iron acquisition pathway involving the function of a conserved family of proteins, including Rbt5, Rbt51/Pga10, Pga7, and Csa2; and (3) the siderophore-mediated uptake pathway employing the Sit1/Arn1 transporter that is able to use xenosiderophores generated by gut or mucosa-colonizing microorganisms (bacteria or other fungi), although studies have shown that similar to *S. cerevisiae*, *C. albicans* also does not synthesize its own siderophores (*Almeida et al., 2009*; *Bairwa et al., 2017*). In addition, our previous studies demonstrated that *C. albicans* has developed a complex and effective regulatory circuit for iron acquisition and storage in host niches, and importantly, the circuit is composed of three iron-responsive transcription regulators Sfu1, Sef1, and Hap43 that control the expression of genes essential for iron uptake and utilization (*Chen et al., 2011*). Recent studies have indicated that Hap43 is required for their response to iron starvation by negatively regulating iron utilization genes (*Blankenship and Mitchell, 2011*; *Chen et al., 2011*; *Hsu et al., 2011*); however, its role in fungal commensalism has never explored. Here, we identified an unexpected regulatory function of this iron-responsive transcription factor that in the iron-rich environment such as GI tract, Hap43 is subject to covalent post-translational modifications (phosphorylation and ubiquitination) leading to cytoplasm-nuclear relocation and proteasome-mediated protein degradation, which serve as a positive signal to de-repress the expression of antioxidant genes. Importantly, we provide sufficient evidence that this regulatory action of Hap43 is most effective in lowering cytotoxicity induced by iron-mediated ROS production and promotes *C. albicans* commensalism in GI tract. Studies from our group and others have put forward a conclusion that *C. albicans* is well equipped to gain access to and utilize iron from host during commensal growth and during both superficial and systemic infection, and evolves sophisticated mechanisms to sense and respond to the fluctuation of iron availability. Given both positive and negative roles for Hap43 in regulating iron homeostasis of *C. albicans*, it would be very interesting to explore the upstream effectors/mechanisms governing the regulatory activity of this transcription factor, for example, the signaling pathways that sense changes in the internal and external concentration of iron, in particular the upstream effectors/regulators that operate in this iron-associated signaling cascade and modulate the expression of Hap43 at transcriptional, post-transcriptional, and post-translational levels.

The host microenvironments differ markedly in the levels of key micronutrients such as iron. For example, the human GI tract is normally recognized as an iron-replete environment because the majority of dietary iron is not absorbed and hypoxia is a strong stimulus for iron absorption could be strongly stimulated by hypoxia and the acidic pH (*McCance and Widdowson, 1938*; *Miret et al.,*

*2003*; *Sharp, 2010*). Interestingly, intestinal absorption of essential trace metals such as iron was found to be highly variable, because the iron contents is strictly regulated by adaptation of the intestinal uptake to the needs of human body, which appear to be affected by a number of factors, including the chemical form, dietary constituents, and the body status (*Diamond et al., 1998*; *Fuqua et al., 2012*). As a consequence, this diversity might instead reflect the iron bioavailability for commensal microbes during their colonization across different regions of the GI tract. As a successful colonizer in GI tract, *C. albicans* may take advantage of the abovementioned mechanisms to obtain sufficient iron to support cellular proliferation and, on the other hand, to protect the fungal cells from iron-mediated toxicity, creating an iron-buffering system that ensures competitive growth of the fungal cells in the fluctuating environment of the GI tract. In addition, the intestinal microenvironment is characterized by variable oxygen levels, for example, the lumen is relatively anaerobic, whereas the region adjacent to the mucosal surface is a zone of relative oxygenation that is generated by diffusion from the microvilli capillary network (*Marteyn et al., 2010*; *Sharp, 2010*). Apparently, *C. albicans* harbors distinct regulatory systems to sense changes in oxygen or iron availability and switch from a commensal to a pathogenic state, the process that is governed by different regulators. The molecular events underlying the intimate crosstalk between iron homeostasis and oxygen metabolism, which requires the activities of different or common effectors/regulators, will be the subject of much investigation in the future.

Protein phosphorylation has been found to affect an estimated one-third of all proteins and recognized as the most widely studied post-translational modification (*Cohen, 2001*). Changes in protein phosphorylation represent an important cell signaling mechanism that is frequently employed by cells to regulate the activities of transcription factors, for example, targeting for proteolytic degradation (*Olsen et al., 2006*). Moreover, a close connection between the ubiquitin-proteasome system and transcriptional activation has been reported in a number of studies (*Lipford and Deshaies, 2003*; *Muratani and Tansey, 2003*). Indeed, studies have shown that the post-translational modification such as the phosphorylation-dependent ubiquitination and degradation is a highly conserved process across eukaryotes. For example, a powerful proteomic study in the budding yeast *S. cerevisiae* identified 466 proteins co-modified with ubiquitylation and phosphorylation (*Swaney et al., 2013*). A variety of extracellular stimuli in mammalian cells cause the rapid phosphorylation, ubiquitination, and ultimately proteolytic degradation of IκB, resulting in cytoplasm-nuclear translocation of NF-κB and induction of gene transcription (*Ghosh and Dass, 2016*). The same is the transcription factor SREBP1 that also undergoes phosphorylation and subsequent ubiquitination and degradation by the proteasome (*Punga et al., 2006*). In *Arabidopsis thaliana*, phosphorylation of the calmodulin-binding transcription activator 3 (CAMTA3) was found to trigger its destabilization and nuclear export (*Jiang et al., 2020*). Consistent with these observations, we described that under iron-replete conditions, the iron-responsive transcription factor Hap43 of *C. albicans* undergoes ubiquitin/proteasome-mediated degradation upon a direct phosphorylation event modulated by Ssn3, a cyclin-dependent kinase previously known to have a similar activity on Ume6 degradation (*Lu et al., 2019*). Although the regions associated with phosphorylation and ubiquitination of Hap43 have been identified in our work, the precise modification sites remain unclear, more likely due to the presence of multiple modification sites and technical challenges such as the detection of low abundant proteins like transcription factors.

Previous studies have shown that ROS production in GI tract could be triggered by different abiotic or biotic stimuli. For example, iron (II) complex was found to interact with bile acids and the K vitamins to generate free radicals in the colon (*Valko et al., 2001*). The host's defense through phagocytes induces an ROS burst that is required for pathogen killing and for regulating pro-inflammatory signaling in phagocytic cells (*El-Benna et al., 2016*). Moreover, similar studies have demonstrated that the antifungal action of different classes of antifungal compounds such as amphotericin B, miconazole, and caspofungin is related with the induction of ROS formation in fungi, especially in *Candida* species (*Mello et al., 2011*). Previous studies showed that miconazole-mediated fungicidal activity against *C. albicans* was significantly inhibited by the addition of antioxidant (*Kobayashi et al., 2002*), and superoxide dismutase inhibitors enhanced the activity of miconazole against *C. albicans* biofilm cells (*Bink et al., 2011*). Our mouse model confirmed that an HFD (400 mg/kg Fe of diet) is sufficient to increase iron levels and ROS generation in mouse colon, which is consistent with previous observation that the same diet is able to increase ROS exposure without being overtly toxic to mice (*Mahalhal et al., 2018*; *Rishi et al., 2018*; *Schwartz et al., 2019*). A surprising observation

from our mouse model is that a normal chow did not increase ROS levels of GI tract and had no effect on intestinal fitness of the mutant lacking Hap43, arguing that the function of this iron responsive regulator might be relevant to either poor diet, a gut disease associated with high levels of ROS, or transient increases in iron availability, or buffering transient increases in iron level to aid commensalism. The ability of *C. albicans* to adapt and respond to oxidative stress is critical for its survival and virulence (*Dantas et al., 2015*). Accumulating evidence has suggested that *C. albicans* cells respond to oxidative stress from the host environment through diverse strategies such as detoxifying ROS, repairing oxidative damages, synthesizing antioxidants, and restoring redox homeostasis, and all of these actions involve the transcriptional induction of antioxidant genes encoding catalase (*CAT1*), superoxide dismutases (*SOD*), glutathione peroxidases (*GPX*), and components of the glutathione/glutaredoxin (*GSH1, TTR1*) and thioredoxin (*TSA1, TRX1, TRR1*) systems (*Mao and Chen, 2019*). Coincidently, the HAP complex, including the Hap31/Hap32/Hap43 subunits, was found to be crucial for the iron-dependent regulation of the oxidative stress response (OSR) in *C. albicans*, especially the activation or repression of OSR genes (*Chakravarti et al., 2017*), and our work also obtained similar results that post-translational modifications of Hap43 effectively modulate the transcription of antioxidant genes in response to iron-induced ROS. In iron-replete environments (e.g. host GI tract), Hap43 degradation leads to de-repression of antioxidant genes that enhances ROS detoxification and promotes the GI colonization of *C. albicans*. Antioxidant enzymes such as superoxide dismutase (SOD) and catalase (CAT) form the first line of defense against ROS in living organisms. Meanwhile, SOD catalyzes the conversion of superoxide into oxygen and hydrogen peroxide and CAT metabolizes the reaction by which hydrogen peroxide is decomposed to oxygen and water (*Ren et al., 2020*; *Van Breusegem et al., 2001*). Our results suggested that both SOD and CAT may largely contribute to the resistance of *HAP43* mutants to iron-induced ROS, which is consistent with a previous study that *C. albicans* Hap43 is the sole Hap4-like subunit responsible for *CAT1* repression (*Chakravarti et al., 2017*).

Therapeutic strategies to tackle iron overload, by the use of iron chelators or antioxidants, have been explored as an adjunct in the treatment of fungal infections, particularly in salvage therapy (*Reed et al., 2006*). Some clinically approved iron-chelating drugs have been directly tested for inhibition of fungal pathogens, including *Cryptococcus, Rhizopus, Candida,* and *Aspergillus* species (*Symeonidis, 2009*). For example, treatment of deferasirox, an FDA-approved iron chelator, significantly decreased the salivary iron levels and *C. albicans* CFUs of tongue tissue in a murine OPC model, and ultimately relieves neutrophil-mediated inflammation (*Puri et al., 2019*). Antioxidant therapy has gradually become popular among patients, largely because of its low cost, less adverse effects, and good treatment outcome (*Trivedi and Jena, 2013*). For example, both COX-2 inhibitors and telmisartan (TLM) are able to inhibit ROS-induced inflammation by upregulating the expression of *GSH* and *GPX* in IBD patients (*Tian et al., 2017*). Sepsis is a systemic inflammatory response induced by an infection (e.g. bacteria or fungi), leading to organ dysfunction and mortality. During sepsis, iron homeostasis becomes disrupted and an excess of ROS is generated, causing damage to tissues. This can be potentially suppressed using iron chelators that selectively bind iron to prevent its participation in ROS-associated inflammatory reactions. Given the importance of Hap43 degradation in ROS detoxification and *C. albicans* commensalism, it is plausible that iron chelator therapy by blocking the process of protein phosphorylation and degradation could be considered as an alternative therapeutic approach against invasive fungal infection. Our findings may deliver new clues for the development of innovative drugs to fight invasive fungal infection.

## Materials and methods

### Media

*C. albicans* strains were grown at 30°C in YPD (1% yeast extract, 2% Bacto peptone, 2% glucose) or SD (0.67% yeast nitrogen base plus 2% dextrose) as 'iron-replete' medium. 'Iron-depleted' medium is YPD or SD supplemented with one of the specific iron chelators, 500 µM bathophenanthroline disulfonic acid (BPS). Doxycycline (50 µg/ml) was added to YPD for Tet-induced expression. When required, MG132 (100 µM) or chloroquine (100 mM) was added to growth medium to inhibit protein degradation.

## Plasmid and strain construction

SC5314 genomic DNA was used as the template for all PCR amplifications of *C. albicans* genes. The *C. albicans* strains used in this study are listed in *Supplementary file 1a*. The primers used for PCR amplification are listed in *Supplementary file 1b*. Plasmids used for Hap43-Myc tagging and knockout gene complementation are listed in *Supplementary file 1c*. Construction of *C. albicans* knockout mutants, complemented strains, strains expressing Myc-tagged Hap43 fusion protein, and overexpression strain for Hap43 was performed as previously described (*Chen et al., 2011*).

To generate tetO-Hap43-Myc strains (CB247), we used the pNIM1 and replaced the caGFP reporter gene by *HAP43*-13xMyc. *HAP43*-13xMyc was amplified with primers (CBO838 and CBO839) that introduced SalI and BglII sites from pSN161. The PCR product was appropriately digested and inserted into SalI/BglII-digested vector pNIM1 to generate pCB127. *hap43Δ/Δ* strain (SN694) was transformed with the following gel-purified, linear SacII-KpnI digested DNA fragments from pCB127. To generate tetO-HA-Ub strain (CB453 and CB494), 3xHA-Ubiquitin (*S. cerevisiae*) was synthesized by company (Gen Script Nanjing Co., Ltd). The plasmid was appropriately digested and inserted into SalI/BglII-digested vector pNIM1 to generate pCB193. Hap43-Myc strain (SN856) or Hap43-Myc, *ssn3Δ/Δ* strain (CB12) was transformed with the following gel-purified, linear SacII- KpnI digested DNA fragments from pCB193, respectively.

## In vitro growth assay

For agar plate assays, fresh overnight yeast cultures were washed and diluted in PBS to adjust the optical density ($OD_{600}$) to 1.0. Then tenfold serial dilutions were prepared and 5 µl aliquots of each dilution were spotted onto appropriate agar plates. For growth curves in liquid media, cells from overnight cultures were diluted to a starting $OD_{600}$ of 0.15 into the indicated medium. At indicated time intervals, optical density at 600 nm ($OD_{600}$) was measured. Presented data (means and SDs) from three technical replicates are shown and plotted in GraphPad Prism.

## Fluorescence microscopy

*C. albicans* was grown at 30°C for 5–6 hr in YPD or YPD supplemented with 500 µM BPS medium to $OD_{600}$ = 0.8–1.0. Cells were fixed by 4.5% formaldehyde for 1 hr and digested by 80 µg/ml Zymolase-20T in 37°C for 15 min. Cells were transformed to polylysine-d-coated culture dishes and remove most of the unattached cells. To flatten cells, add pre-cold (–20°C) methanol for 5 min followed by precold (–20°C) acetone for 30 s. The 9E10 anti-c-Myc antibody was used at a 1:150 dilution overnight after cells were completely dry. A 1:400 dilution of Cy2-conjugated secondary antibody was used for 1 hr. Images were acquired under oil objective using the inverted microscope (Olympus IX73) or high-resolution confocal microscope (Olympus FV-1200). DIC, DAPI, and FITC images acquired.

## Promoter shutdown assays

*C. albicans* strains containing Hap43-Myc or 3xHA-Ubiquitin under the regulation of the tetO promoter were grown in YPD at 30°C overnight, then diluted 1:100 into YPD plus 50 µg/ml DOX to induce the expression of Hap43-Myc. Then the medium was replaced by fresh YPD medium at 30°C to shut off the promoter. Aliquots were collected after the times indicated.

## Protein extraction and immunoblotting

*C. albicans* protein extracts were prepared under denaturing conditions. Briefly, lysates corresponding to 1 $OD_{600}$ of cells were analyzed by SDS-PAGE and immunoblotted with either anti-c-Myc (9E10, Covance Research) for Myc-tagged proteins or anti-peroxidase soluble complex antibody (Sigma, P2416) for TAP-tagged proteins. Immunoblots were also probed with anti-alpha tubulin antibody (Novus Biologicals, NB100-1639) as a loading control. At least three biological replicates were obtained for each experiment shown, and ImageJ software was used for densitometry.

## Lambda phosphatase treatment

100 ml *C. albicans* cells in log phage was washed with 1 ml ice-cold 1.2 M sorbitol twice and split into two tubes. Add 500 µl protein extraction buffer (420 mM NaCl, 200 µM EDTA, 1.5 mM MgCl$_2$, 10% glycerol, 0.05% Tween-20, 50 mM Tris-Cl, pH 7.5) containing 0.5 M fresh-made DTT and protease inhibitor cocktails (Roche, USA). Add 0.5 mm glass beads and break cells by vortex (6 × 30 s, 2 min

interval on ice, top speed, 4°C). Spin at top speed and transfer supernatant to a new tube. Add 5 µl 10× PMP buffer, 5 µl 10 mM $MnCl_2$ buffer and 0 or 2 µl lambda phosphatase (NEB #P0753S, USA) in 38 µl supernatant. The mix was incubated for 60 min at 30°C, followed by 10 min in 65°C.

## Immunoprecipitation and pull-down assay

100 ml cells expressing TAP-tagged Hap43 or Ssn3 as well as cells expressing HA-tagged ubiquitin were collected by centrifugation in log phage. Cells were washed three times with ice-old water and resuspended in 1 ml of lysis buffer (20 mM Tris, pH 7.4, 100 mM KCl, 5 mM $MgCl_2$, 20% glycerol) with protease and phosphatase inhibitors (Roche). Cells were lysed using a Bead Beater and 300 µl of glass beads. Cell lysates were centrifuged for at max speed at 4°C for 15 min. Protein concentration of the supernatants was measured by the Bradford assay and whole-cell extracts were collected in freezer. 3 mg of proteins was used for immunoprecipitation with 50 ml of immunoglobulin G-Sepharose resin (IgG Sepharose 6 Fast Flow, GE Healthcare) or Anti-HA affinity matrix beads (Roche). After protein overnight rotation at 4°C, the resin was washed three times with lysis buffer. For TAP-tagged proteins, the resin was washed twice with tobacco etch virus (TEV) protease cleavage buffer (10 mM Tris-HCl, pH 8, 150 mM NaCl, 0.5 mM EDTA, 0.1% Tween-20). Halo TEV protease (Promega, USA) cleavage was performed in 1 ml buffer at 4°C overnight. The TEV eluate was collected and proteins were recovered by trichloroacetic acid (TCA) precipitation. For HA-tagged proteins, the resin was boiled in SDS-PAGE loading buffer (50 mM Tris-HCl, pH 6.8, 2% SDS, 10% [v/v] glycerol, 2 mM DTT, 0.01% [w/v] bromophenol blue).

## Nuclear fraction separation

The nuclear fraction was prepared as a described method (*von Hagen and Michelsen, 2013*). 50 ml *C. albicans* cells in log phage was washed with preincubation buffer (100 mM PIPES-KOH pH 9.4, 10 mM DTT). The pellet was resuspended preincubation buffer and incubate in 30°C for 10 min. Spin down and resuspend in 2 ml lysis buffer (50 mM Tris-HCl pH 7.5, 10 mM $MgCl_2$, 1.2 M sorbitol, 1 mM DTT) plus 80 µl Zymolase 20T (2.5 mg/ml) and incubate in 30°C for 60 min. Cells were centrifuged and washes in lysis buffer twice and were resuspended in 2 ml Ficoll buffer (18% Ficoll 400, 100 mM Tris-HCl, pH 7.5, 20 mM KCl, 5 mM $MgCl_2$, 3 mM DTT, 1 mM EDTA) containing protease inhibitors as described above. The cells were lysed using a Dounce homogenizer. Unlysed cells and cell debris were removed by 3000 rpm 15 min centrifugation. The supernatant was equally divided in two portions: one was saved as whole-cell extract and the other was centrifuged at max speed for 15 min. The pellet (nuclear) was washed by PBS twice and resuspended in SDS loading buffer.

## Fungal genomic DNA isolation and total RNA preparation for RT-qPCR

Fungal genomic DNA isolation was performed as previously described (*Chen et al., 2012*). Samples were harvested by centrifugation, and the pellets were resuspended in a DNA extraction solution containing 200 µl of breaking buffer (2% Triton X-100, 1% SDS, 100 mM NaCl, 10 mM Tris-HCl pH 8.0, 1 mM EDTA pH 8.0), 200 µl of acid phenol:chloroform:isoamyl alcohol (pH 8.0, Ambion) and a slurry of acid-washed glass beads (Sigma-Aldrich). Fungal cells were mechanically disrupted with a FastPrep-24 5G (MP Biomedicals, USA), and genomic DNA were extracted and precipitated with isopropanol.

Fungal RNA was prepared as described previously (*Chen et al., 2012*), 1–2 µg of each RNA was treated with RNase-free Dnase I (Promega, Madison, WI) and reverse transcribed using the Prime-Script RT reagent Kit (TaKaRa). qPCR was performed using the SYBR Green Master Mix (High ROX Premixed; Vazyme, Nanjing, China) using the primers in *Supplementary file 1b*. Normalization of expression levels was carried out using the *ACT1* genes, and the primers for *ACT1* were used as previously described. At least three biological replicates were performed per strain per condition.

## Chromatin immunoprecipitation

ChIP experiments were performed essentially as described (*Nobile et al., 2009*). Unless otherwise noted, cells were crosslinked with 1% formaldehyde for 20 min at 30°C, followed by 125 mM glycine for 5 min. Cell pellets were resuspended in 700 ul ice-cold lysis buffer (50 mM HEPES-KOH pH 7.5, 140 mM NaCl, 1 mM EDTA, 1% Triton X 100, 0.1% NaDOC) with protease inhibitor cocktails (Roche). Vortex with 300 ul glass beads at max speed for ~2 hr at 4°C. Recover the lysate by inverting and centrifuging the tubes with punctures on bottom and top of tubes by a 26G needle. Hap43-Myc were

immunoprecipitated with 2–5 mg antibody (anti-Myc, 9E10, Covance Research) from lysates corresponding to optical density 600 ($OD_{600}$) of cells at 4°C overnight. Add 50 μl of prepared A or G beads suspension to each IP sample and rotate for 2 hr at 4°C. Wash twice with lysis buffer, high salt lysis buffer (50 mM HEPES-KOH pH 7.5, 500 mM NaCl, 1 mM EDTA, 1% Triton X 100, 0.1% NaDOC), and wash buffer (10 mM Tris-Cl pH 8.0, 250 mM LiCl, 0.5% NP-40, 0.5% NaDOC, 1 mM EDTA), respectively, and resuspend in TE buffer. Products were eluted in elution buffer and incubated in 65°C. DNA was de-crosslinked by proteinase K (Sigma, USA) and 4 M LiCl and purified by phenol:chloroform:isoamyl alcohol. Immunoprecipitated DNA was quantified by real-time PCR (qPCR) with primers and normalized against *ACT1*.

## Measurement of ROS production

3′(4-Hydroxyphenyl)-fluorescein (HPF; Molecular Probes, OR) was used for detecting •OH production (*Avci et al., 2016*). Log-phased *C. albicans* cells were washed in PBS buffer twice. HPF fluorescent probe was added to washed cells and kept in a 37°C shaker for 30 min. Subsequently, cells were centrifuged (3200 rpm, 3 min) immediately and were resuspended in PBS. The stained cells were detected by a fluorescent microplate reader (Thermo Fisher, USA) or BD LSR Fortessa flow cytometer (BD Bioscience). Cells were also counted using a hemocytometer. The relative fluorescence density of each sample was calculated as FLU divided by the cell number to evaluate intracellular ROS levels.

## Determination of colonic $H_2O_2$

Determination of $H_2O_2$ was performed according to the protocol of Beyotime kit (Cat #S0038, Beyotime, China). Briefly, 50 mg of colon tissue fragment was homogenized with 200 ul of lysis buffer and was centrifuged at max speed for 5 min in 4°C. Subsequently, 50 μl of supernatant was mixed with 100 ul test buffer and incubate for 30 min in room temperature. Then A560 was detected by a fluorescent microplate reader (Thermo Fisher, USA). Readings were calculated by the standard curve that was prepared from three series of calibration experiments with five increasing $H_2O_2$ concentrations (ranging from 1 to 100 μM/l).

## Mice infection assays

Female C57BL/6 mice (6–8 weeks old, weighing 18–20 g) were purchased from Beijing Vital River Laboratory Animal Technology Company (Beijing, China). The mice were routinely maintained in a pathogen-free animal facility at Institut Pasteur of Shanghai, Chinese Academy of Sciences. All mice had free access to food and water in a specific pathogen-free animal facility with controlled temperature, humidity, and a preset dark-light cycle (12 hr:12 hr). Infections were performed under SPF conditions. The 'normal diet' was a standard chow diet (37 mg/kg iron; Shanghai SLAC Laboratory Animal Co., Ltd). The 'high-iron diet' was a diet supplemented with 400 mg/kg iron (Shanghai SLAC Laboratory Animal Co., Ltd). For competed infection, mice were received penicillin (1.5 mg/ml) and streptomycin (2 mg/ml) in their drinking water for 3 d prior to gavage with $1 \times 10^8$ CFUs of 1:1 mixtures. Stool samples were homogenized in PBS and cultured in Sabouraud plates supplemented with ampicillin 50 μg/ml and gentamicin 15 μg/ml. Genome DNA were was extracted for fitness value of each strain by qPCR using strain-specific primers. For individual infection, mice were colonized as above but with a single strain totaling $1 \times 10^8$ CFUs/mouse. Stool samples were collected at indicated days post-inoculation, homogenized in PBS, diluted and plated on Sabouraud agar medium containing ampicillin 50 μg/ml and gentamicin 15 μg/ml. Colonization was measured by counting the number of *C. albicans* cells grown on each plate. For iron staining, colons were fixed with 10% formalin, and paraffin-embedded sections were stained with fresh-made Prussian blue staining solution. For ROS staining, colons were 'snap-frozen' in optimum cutting temperature compound, and frozen sections were stained with DHE and DAPI. For colonic RNA, RNA was extracted with TRIzol according to the manufacturer's instructions (Invitrogen). For periodic acid-Schiff (PAS) and immunofluorescent staining, visualization of *C. albicans* cells in GI tract was performed as described previously (*Wang et al., 2022*). For NAC treatment, drinking water containing 1.5 g/l NAC were given to the mice during the course of experiments.

## Statistical analysis

Data are presented as mean ± SD for continuous variables. All statistical analyses were performed with GraphPad Prism 8 (San Diego, CA), and details are provided in the figure legends. The following p-values were considered: *p<0.05; **p<0.01; ***p<0.001; ****p<0.0001.

## Acknowledgements

CC is supported by the grant from the MOST Key R&D Program of China (2022YFC2303200); the National Natural Science Foundation (32170195); the Shanghai Municipal Science and Technology Major Project (2019SHZDZX02); the Open Project of the State Key Laboratory of Trauma, Burns and Combined Injury, Third Military Medical University (SKLKF201803); LZ is supported by the grant from the MOST Key R&D Program of China (2022YFC2304700); HL is supported by the Special Program of PLA (19SWAQ18). YW is supported by a National Postdoctoral fellowship (No. 312780) and Shanghai Post-doctoral Excellence Program (No. 2022647). XH is supported by grants from the National Natural Science Foundation (32070146, 31600119); the Natural Science Foundation of Shanghai (20ZR1463800, 15ZR1444400). The authors specially thank Dr. Suzanne Noble at UCSF for providing us the *C. albicans* homozygous gene deletion library and related plasmids as gifts, a powerful research platform for fungal studies, and excellent supervision in postdoctoral education and training. We gratefully acknowledge Dr. Aaron P Mitchell at the University of Georgia for valuable comments and helpful discussions. We thank all the lab members in the Chen-lab at Institut Pasteur of Shanghai, Chinese Academy of Sciences, the members in the Liang-lab at State Key Laboratory of Trauma, Burns and Combined Injury, Third Military Medical University, and the members in the Zhong-lab at Department of General Surgery, Shanghai General Hospital, Shanghai Jiao Tong University School of Medicine, for their help in discussion and preparation of the manuscript.

## Additional information

### Funding

| Funder | Grant reference number | Author |
|---|---|---|
| Ministry of Science and Technology (China) | Key R&D Program 2022YFC2303200 | Changbin Chen |
| Ministry of Science and Technology (China) | Key R&D Program 2022YFC2304700 | Lin Zhong |
| National Natural Science Foundation of China | 32170195 | Changbin Chen |
| The Shanghai Municipal Science and Technology Major Project | 2019SHZDZX02 | Changbin Chen |
| The Open Project of the State Key Laboratory of Trauma, Burns and Combined Injury, Third Military Medical University | SKLKF201803 | Changbin Chen |
| The Special Program of PLA | 19SWAQ18 | Huaping Liang |
| National Postdoctoral Fellowship | No.312780 | Yuanyuan Wang |
| Shanghai Post-doctoral Excellence Program | No.2022647 | Yuanyuan Wang |
| National Natural Science Foundation of China | 32070146 | Xinhuang Huang |
| National Natural Science Foundation of China | 31600119 | Xinhuang Huang |

| Funder | Grant reference number | Author |
|---|---|---|
| Natural Science Foundation of Shanghai | 20ZR1463800 | Xinhuang Huang |
| Natural Science Foundation of Shanghai | 15ZR1444400 | Xinhuang Huang |

The funders had no role in study design, data collection and interpretation, or the decision to submit the work for publication.

## Author contributions

Yuanyuan Wang, Data curation, Funding acquisition, Investigation, Writing – original draft, Project administration; Yinhe Mao, Data curation, Investigation, Methodology, Writing – original draft, Project administration; Xiaoqing Chen, Software, Visualization, Methodology, Project administration; Xinhuang Huang, Funding acquisition, Methodology, Project administration; Zhongyi Jiang, Investigation, Methodology; Kaiyan Yang, Lixing Tian, Xiaoyuan Ma, Chaoyue Xu, Zili Zhou, Methodology; Tong Jiang, Xianwei Wu, Supervision, Methodology; Yun Zou, Supervision, Investigation, Methodology; Lei Pan, Resources, Methodology; Huaping Liang, Changbin Chen, Conceptualization, Funding acquisition, Writing - review and editing; Lin Zhong, Conceptualization, Funding acquisition, Methodology, Writing - review and editing

## Author ORCIDs

Yuanyuan Wang ⓘ http://orcid.org/0000-0001-7464-7446
Changbin Chen ⓘ http://orcid.org/0000-0002-7961-3488

## Ethics

All animal experiments were carried out in strict accordance with the regulations in the Guide for the Care and Use of Laboratory Animals issued by the Ministry of Science and Technology of the People's Republic of China. All efforts were made to minimize suffering. The protocol was approved by IACUC at the Institut Pasteur of Shanghai, Chinese Academy of Sciences (Permit Number: A160291).

## Decision letter and Author response

Decision letter https://doi.org/10.7554/eLife.86075.sa1
Author response https://doi.org/10.7554/eLife.86075.sa2

# Additional files

## Supplementary files

- Supplementary file 1. Strains (a), plasmids (b), and primers (c) used in this study.
- MDAR checklist

## Data availability

All data generated or analysed during this study are included in the manuscript and supporting file. Source data files have been provided for all figures.

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
