## [Editor Report]

This important study reveals a novel mechanism that links iron stress to anti-oxidant protection for the gut commensal and opportunistic pathogen *Candida albicans* in colonization of the gastrointestinal tract. Through a series of convincing experiments, the authors show that phosphorylation and degradation of Hap43, a well-established regulator of *Candida albicans* iron homeostasis, underlies the interaction between this gut commensal and the mammalian gut. The work will be of interest to microbiologists working on microbiota and microbial commensalism.

---

## [Decision Letter]

**Decision letter after peer review:**

Thank you for submitting your article "Homeostatic control of an iron repressor in a GI tract resident" for consideration by *eLife*. Your article has been reviewed by 3 peer reviewers, and the evaluation has been overseen by a Reviewing Editor and Wendy Garrett as the Senior Editor. The following individuals involved in review of your submission have agreed to reveal their identity: Yu-Huan Tsai (Reviewer #1); Campbell Gourlay Gourlay (Reviewer #2); Sascha Brunke (Reviewer #3).

Essential revisions:

1) All three reviewers made a series of constructive suggestions that will improve the quality and clarity of your manuscript. In particular, please pay attention to reviewers' comments suggesting that some of the conclusions drawn in the manuscript require more experiments or rephrasing. We suggest that the authors rephrase these conclusions to the extend that they are supported by the findings.

2) There was consensus among reviewers that the experiment on the influence of iron in the *Drosophila* experiment was unclear and could be dropped.

*Reviewer #1 (Recommendations for the authors):*

In this study, the authors provide the evidence that in *Candida albicans*, excess ROS brought by environmental iron would lead to degradation of the iron-responsive transcription regulator Hap43 dependent on multiple potential phosphorylation sites on this protein, which leads to de-repression of antioxidant genes directly repressed by Hap43. This newly identified molecular mechanism implicating the post-translational degradation of Hap43 in *C. albicans* may contribute to ROS detoxification and higher gut colonization of this fungal commensal.

The biochemical data regarding the implication of Hap43 phosphorylation in its degradation by applying the ssn3 kinase mutant and substitutions of all 29 potential phosphorylation sites are compelling and convincing. The results in Figure 5 also provide solid evidence to discuss the interplay between environmental iron and ROS in controlling Hap43 degradation. However, the data regarding the four phosphorylation sites in the short list are incomplete. For example, while the HAP43tr and HAP43m29 mutants show decreased growth in the presence of ROS due to their resistance to Hap43 degradation as expected, the HAP43m4 mutant does not have any detectable phenotype when growing with ROS. This is possibly due to potential implication of additional phosphorylation sites between the amino acid 400 to 504, as there was another drop between delta400 and delta504 in Hap43-Myc degradation screening assay in Figure 4C. Moreover, the authors claimed in the manuscript that the degradation of Hap43 depends on its ubiquitination, while the authors only showed the capacity of Hap43 to be ubiquitinated in the presence of ssn3. It is better to tone down the conclusion without further experimental evidence. Another potential misleading point is that the author described the shift of Hap43-Myc to a larger size in Western blots as "a rapid gel mobility" or "an increase in the electrophoretic mobility". However, the shifted up is supposed to be due to a decrease in the electrophoretic mobility.

Concerning the animal experiments, the authors used two different models, mice and flies, to show the advantage of HAP43 deletion in the gut, although caution has to be exercised in interpreting the 2 to 3-fold difference in the advantage. Moreover, while the authors discussed that the high-Fe diet (HFD) would not be overtly toxic to mice, it is not clear whether HFD can induce *C. albicans* tissue infection during colonization. In line with this, in the fly infection model, *C. albicans* can cause extensive cell death short after ingestion (Glittenberg et al., 2011). The increased fitness of HAP43 KO mutant could be due to their higher resistance to tissue inflammation concerning ROS. While the title is still relevant in this notion, it is better not to only focus on gut commensalism but to expand the conclusion to the interaction between *C. albicans* and the gut, where infection might be involved.

Altogether, despite some overstatements in interpretation of the results, this finding provides a novel point of view discussing the potential damage brought by excess environmental iron, in contrast to the traditional view of taking iron as an important nutrient for microbes, when living in a host.

In general, if you have shortlisted the phosphorylation sites to the four demonstrated in Figure 4—figure supplement 5, would it be possible to show the results by HAP43m4 in the main text and move those with HAP43tru and HAPm29 to the supplement? In this case, it would be more convincing to include the experimental results in Figure 5G, 5H and 6 using HAP43m4. This would support the conclusion regarding the implication of Hap43 phosphorylation in its degradation. Otherwise, the data with HAP43tru can only claim the Importance of the truncated region.

L140. Suggest"s".

Figure 1B. It appears that the ROS was largely detected in the tissues but not in the lumen where microbes are mostly located. Would be possible to specifically detect ROS level in the lumen in a more specific manner?

Figure 1G and 1H. Would the HFD make the mice susceptible to *C. albicans* intestinal tissue invasion, where HAP43 has a role? This can be achieved by immunofluorescence staining of the tissues from HFD mice inoculated by *C. albicans*. Can treatment of antioxidant abrogate the fitness of HAP43 mutant in HFD?

L190-194. Do flies have higher ROS in the gut in nature so that there was no need to provide excess iron to show the advantage of HAP43 deletion? If yes, better to provide relevant reference for the iron/ROS issues in flies.

L203. The review cited here discussed that duodenum can uptake iron. Is this what you wanted to highlight?

Figure 3A. Would it be possible to use confocal microscope to confirm the co-localization? If SSN3-mediated Hap43 phosphorylation results in Hap43 degradation, would one anticipate a sharp decrease in fluorescence intensity instead of restricted localization in cytosol with comparable fluorescence intensity?

Figure 3B. Do you expect to see a shift of Hap43-Myc between high and low iron in WT as in figure 2B? Does this result also demonstrate the potential implication of another kinase for Hap43 phosphorylation as most of Hap43 proteins left the nucleus in iron replete condition in ssn3 deletion mutant?

L254-256. At this point we can only say that Hap43 size shift and cellular localization can be modulated by SSN3. We do not even know whether Hap43 can be phosphorylated by SSN3, which may require purified proteins to prove.

L256-258. Current evidence only supports that Hap43 expression and localization depends on SSN3 expression.

L275-276. There is no data supporting the dependence of Hap43 phosphorylation by SSN3. Hap43 phosphorylation have never been demonstrated.

Figure 3—figure supplement 1. Do you expect that the dynamic of Hap43-Myc degradation was much similar to the condition with MG132 in figure 3E?

Figure 5C. Is it the time after Hap43-Myc overexpression? Better to describe more clearly in the figure legends.

Figure 5D. Why did you have Hap43-Myc expression 120 min after induction here, but not in figure 4C in YPD only culture condition?

L409. The current data do not demonstrate the ubiquitination-dependent degradation Hap43, but just show SSN3-dependent ubiquitination (Figure 3F) and phosphorylation-dependent degradation (Figure 4F), respectively.

L478-483. No further data to support the conclusion. I would suggest to move this part to discussion.

*Reviewer #2 (Recommendations for the authors):*

The authors present a significant body of work that demonstrates a novel mechanism by which the human commensal yeast *Candida albicans* can adapt to high levels of iron in the GI tract. The paper includes a detailed set of carefully conducted experiments in vitro and in vivo that show a mechanisms that links high iron levels to the management of reactive oxygen species levels in *C. albicans* cells. This mechanism provides a compelling argument for a strategy that will allow colonisation of the GI tract. The mechanism also represents a potential route to manage *C. albicans* colonisation of the GI tract, an issue that is increasingly linked to a number of gut disorders.

This is an excellent study that presents a novel mechanism that allows *C. albicans* to thrive in an iron rich environment such as the GI tract. It is an elegant study and I had very few issues with the data presented, which were compelling, well delivered and of very high quality. I do not believe that additional experiments are required as the paper does lead a significant advance in the field.

Some points to consider that I think would improve the paper are as follows:

Abstract – some grammatical errors are present, please review.

Importance – "iron homeostasis is critical for creatures"?? this does not read as a scientific statement, please adjust.

The importance summary requires revision to correct a number of many grammatical errors. It also does not really convey the importance of this research, why is it of interest?

Results – figure 1 – line 164 change "neglectable" to negligible.

The adjustment of diet to 400mg/Kg to ensure that Ros levels were increased needs to be properly discussed, as in a normal diet HAP43 was not required, the results are therefore relevant to either poor diet, a gut disease associated with high ROS or transient increases in iron associated, or is it that Hap43 is required to buffer transient increases in iron level to aid commensalism? Please include a discussion of this.

Figure 6 It was curious that TRR1 levels were not altered despite binding showing a significant increase, this should be discussed.

The discussion itself is rather brief, it would be useful to discuss how the iron may be absorbed by Candida cells as it does not have, as yet, identified siderophores. There is evidence for several iron uptake mechanisms and this should be mentioned. Although I do not suggest carrying out more experiments to expand the mechanism, presumably identifying the upstream mechanism would be the next step? and this should be discussed. It would also be important to discuss this mechanism within the context of the GI tract, which presumably has highly variable levels of iron. Is the system a buffer to ensure that the cells can remain competitive in the fluctuating environment of the GI tract for example? how would the system interplay with variable oxygen levels found in the GI tract?

*Reviewer #3 (Recommendations for the authors):*

In this work, the authors show a role in gut colonization for the well-established regulator of *Candida albicans* iron homeostasis, Hap43. They link an increased ability of a Hap43 deletion mutant to colonize the gut of mice under a high-iron diet to the repressor role of Hap43 in oxidative stress response (and the iron-dependent Fenton reaction that creates such stress). They go on to show the interplay between posttranslational modification, localization, degradation and gene regulation by Hap43 in response to shifting iron levels.

Overall, there are some very interesting findings in this manuscript, and it definitely adds to our understanding of iron-dependent regulation in *C. albicans*. The set of experiments is rather comprehensive and supports the conclusions. It will certainly be of interest to scientists in the field of Candida and fungal pathogenesis in general, as well as those interested in the role of micronutrients in host-pathogen interactions.

Experiment-wise, I was fully convinced by the authors' systematic approach, and all the conclusions seem to be fully supported by the data.

My main concern is that some aspects are not as novel as depicted in the text (and the relevant literature is missing from the cited references). For example, the regulation of the oxidative stress response in dependence on iron (and especially Hap43) has been described in Chakravarti et al. (2016). This is implied, but not stated, in the beginning (around line 390), there citing a review instead. However, later (line 550) the text states that "… we discovered the role of the transcription factor Hap43 in modulation of the transcription of antioxidant genes in response to iron". I believe this connection has been known before?

Similarly, while it was interesting to see multiple phosphorylation sites in Hap43, the observation of phosphorylation and subsequent degradation of Hap43 was not that surprising. Essentially the same has been shown before for its A. fumigatus homologue, HapX, complete with the role of the proteasome in this process (in the similarly uncited Lopez-Berges et al. (2021)). In the present manuscript, the authors show much more data, including the localization and the responsible kinase, but the data should be put into context of the Aspergillus HapX findings (e.g. with the non-phosphorylatable mutant tested there or the identical effect of MG132).

The connection to the diet is interesting, although a bit more critical discussion may be in order. How realistic are these conditions in the human colonization situation -- the Fe input is higher than the highest concentration given for human feces in the introduction? And why did the same effect of the Hap43 mutant occurs in the *Drosophila* model, although it does not seem to be especially iron-rich?

The sections on importance (and to some extent, the abstract) are in need of some proofreading/rewriting -- e.g. "in the GI tract where is iron-replete environment", "the mystery in iron-replete conditions of Hap43 has never been uncovered" and so on. This is strikingly different to the rest of the text, which is quite easy to follow and rather well-written.

---

## [Author Response]

Essential revisions:1) All three reviewers made a series of constructive suggestions that will improve the quality and clarity of your manuscript. In particular, please pay attention to reviewers' comments suggesting that some of the conclusions drawn in the manuscript require more experiments or rephrasing. We suggest that the authors rephrase these conclusions to the extend that they are supported by the findings.

We are very grateful for the valuable and constructive suggestions from the reviewers. According to these advices, we have performed several new experiments and revised the manuscript. We believe that the manuscript is substantially improved after making the suggested edits.

2) There was consensus among reviewers that the experiment on the influence of iron in the *Drosophila* experiment was unclear and could be dropped.

We accepted the reviewers’ suggestions and removed the *Drosophila* parts in the revised manuscript, in order to avoid possible misleading information.

Reviewer #1 (Recommendations for the authors):In this study, the authors provide the evidence that in Candida albicans, excess ROS brought by environmental iron would lead to degradation of the iron-responsive transcription regulator Hap43 dependent on multiple potential phosphorylation sites on this protein, which leads to de-repression of antioxidant genes directly repressed by Hap43. This newly identified molecular mechanism implicating the post-translational degradation of Hap43 in C. albicans may contribute to ROS detoxification and higher gut colonization of this fungal commensal.[…]In general, if you have shortlisted the phosphorylation sites to the four demonstrated in Figure 4—figure supplement 5, would it be possible to show the results by HAP43m4 in the main text and move those with HAP43tru and HAPm29 to the supplement? In this case, it would be more convincing to include the experimental results in Figure 5G, 5H and 6 using HAP43m4. This would support the conclusion regarding the implication of Hap43 phosphorylation in its degradation. Otherwise, the data with HAP43tru can only claim the Importance of the truncated region.

We are grateful to the reviewer for these constructive suggestions. We analyzed the cellular localization of Hap43m4 through indirect immunofluorescence assay, expression levels of antioxidant genes by qPCR, and commensal fitness of Hap43m4 mutant in GI tract of mice. As shown in Figure 5 and Figure 6—figure supplement 2, we found that similar to both Hap43m29 and Hap43tr mutants, the Hap43m4 mutant generated by amino acid replacement of the four putative phosphorylation sites (S337/S355/S369/T381), also displayed nuclear relocation of Hap43 (Figure 6—figure supplement 2A), decreased expression of antioxidant genes (Figure 6—figure supplement 2B) and decreased commensal fitness in GI tract (Figure 5J). These data further support our conclusion that iron-induced Hap43 phosphorylation, followed by ubiquitin-dependent proteasomal degradation, acts to protect *C. albicans* from ROS toxicity and thus promote its survival in GI tract.

Of course, compared to the Hap43m29 and Hap43tr mutants, the Hap43m4 mutant exhibits relatively weak phenotypes, as we observed (Figure 5J and Figure 6—figure supplement 2) that the mutated Hap43 protein relocates to nucleus at lower levels (~80% vs ~40%), the expression levels of antioxidant genes show less marked changes, and the commensal fitness decreased only slightly but significantly. The differences could be due to involvement of some other uncharacterized phosphorylation sites. Based on the results of Figure 4, we could not exclude the contribution of other phosphorylation sites in conveying the signal for protein degradation, in addition of the tested four sites (S337/S355/S369/T381).

In the revised manuscript, Figure 5 was re-organized based on the reviewer’s suggestions. We removed the fly infection results and included all in vivo evidence showing that the *HAP43* mutant-29, Hap43 truncation and *HAP43* mutant-4 mutants could be outcompeted by the WT strain when cells stably colonize in mouse GI tract (Figure 5H-J). Figure 6 was not changed because it is difficult to put so many results from the three mutants into single figure and the Hap43m4 mutant exhibits relatively weak phenotypes when compared to the Hap43m29 and Hap43tr mutants.

L140. Suggest"s".

We have corrected it in the revised manuscript. (Line 143 on page 8)

Figure 1B. It appears that the ROS was largely detected in the tissues but not in the lumen where microbes are mostly located. Would be possible to specifically detect ROS level in the lumen in a more specific manner?

We thank the reviewer for this pertinent suggestion. A recently published paper by Iliyan D. Iliev group (Cell, 2022) suggested that *Candida* species appear to be mainly located at the intestinal mucosa but not lumen. As described in the paper, the luminal-associated mycobiota included genera that were likely transient-environmental organisms such as *Cladosporium* and *Aspergillus* (Hallen-Adams and Suhr, 2017), whereas the mucosa-enriched operational taxonomic units (OTUs) belonged to several ‘‘immunoreactive’’ fungal genera, such as *Candida*, *Saccharomyces*, and *Saccharomycopsis*. A similar mycobiota composition was observed in human subjects where *Candida spp*. and *Saccharomyces spp*. were the dominant genera associated with the intestinal mucosa of surgical specimens. Therefore, we focused on the mucosa-associated intestine tissue which is a major source of ROS.

References:

Leonardi I *et al.* (2022) Mucosal fungi promote gut barrier function and social behavior via Type 17 immunity. *Cell* 185(5): 831-846.

Hallen-Adams HE and Suhr MJ. (2017) Fungi in the healthy human gastrointestinal tract. *Virulence* 8(3): 352-358.

Figure 1G and 1H. Would the HFD make the mice susceptible to C. albicans intestinal tissue invasion, where HAP43 has a role? This can be achieved by immunofluorescence staining of the tissues from HFD mice inoculated by C. albicans. Can treatment of antioxidant abrogate the fitness of HAP43 mutant in HFD?

We appreciate the reviewer’s constructive suggestions. First, we set up a new experiment using our commensal model to test whether HFD could affect the intestinal colonization of *C. albicans*. Notably, starting from day 5 post-inoculation, we observed a significant increase of fungal loads in feces of HFD-fed mice, compared to that of NFD-fed mice (Figure 1—figure supplement 2A). Consistently, both periodic acid-Schiff (PAS) and immunofluorescent staining (anti-*Candida* antibody) detected a significantly higher enrichment of fungal cells in colon of HFD-fed mice (Figure 1—figure supplement 2B). The results indicate that HFD indeed makes the mice more susceptible to *C. albicans* colonization in GI tract whether Hap43 has a role in modulating fungal fitness. In support of our conclusion, a recent literature reported that the use of the iron chelator deferasirox is able to decrease *C. albicans* invasion of oral epithelial cells and infection levels in murine oropharyngeal candidiasis. (Lines 187-194 on pages 10-11)

Second, we also examined the effect of antioxidant (NAC) on intestinal colonization of Hap43 mutant in HFD-fed mice. Our results showed that NAC treatment is able to partially but significantly abrogate the commensal fitness of Hap43 mutant in HFD-fed mice (Figure 1H). (Lines 199-201 on page 11)

Reference:

Puri S *et al.* (2019) Iron Chelator Deferasirox Reduces *Candida albicans* Invasion of Oral Epithelial Cells and Infection Levels in Murine Oropharyngeal Candidiasis. *Antimicrob Agents and Chemother* 63(4): e02152-18.

L190-194. Do flies have higher ROS in the gut in nature so that there was no need to provide excess iron to show the advantage of HAP43 deletion? If yes, better to provide relevant reference for the iron/ROS issues in flies.

We accepted the reviewer’s comments and removed the *Drosophila* parts in the revised manuscript, in order to avoid possible misleading information.

L203. The review cited here discussed that duodenum can uptake iron. Is this what you wanted to highlight?

We apologize for the lack of clarity. This sentence has been rephrased in our revised manuscript. Here we want to emphasize that commensal microbes colonized in the GI tract are thrived in comparatively high levels of iron because the majority of dietary iron is not absorbed, based on previous reports (McCance and Widdowson, 1938; Miret et al., 2003). (Lines 205-207 on page 11)

References：

McCance RA and Widdowson EM (1938) The absorption and excretion of iron following oral and intravenous administration. *J Physiol.* 94(1):148-54.

Miret S *et al.* (2003) Physiology and molecular biology of dietary iron absorption. *Annu Rev Nutr.* 23: 283-301.

Figure 3A. Would it be possible to use confocal microscope to confirm the co-localization? If SSN3-mediated Hap43 phosphorylation results in Hap43 degradation, would one anticipate a sharp decrease in fluorescence intensity instead of restricted localization in cytosol with comparable fluorescence intensity?

We thank the reviewer for valuable suggestions. We repeated our indirect immunofluorescence assay using high-resolution confocal microscope and obtained similar results that Hap43-Myc was partially mislocalized from cytoplasm to nucleus in *ssn3Δ/Δ* mutant, compared to a complete cytoplasmic localization of this fusion protein in WT (Author response image 1). In addition, we quantified the fluorescence in images from staining samples obtained by confocal microscopy and did observe a significant increase in fluorescence intensity in *ssn3Δ/Δ* mutant when compared to that in WT (Author response image 1).

**Author response image 1. sa2fig1:** (A) Left panels: Indirect immunofluorescence of Hap43-Myc in WT and *ssn3Δ/Δ* mutant strains grown under iron-replete conditions. DIC represents phase images, DAPI represents nuclear staining, FITC represents Hap43-Myc staining, and Merge represents the overlay of Hap43-Myc and nuclear staining. Right panels: Quantification of the cellular distribution of Hap43. Each bar represents the analysis of at least 100 cells. C representing >90% cytoplasmic staining, N >90% nuclear staining, and C+N a mixture of cytoplasmic and nuclear staining. Scale bar, 5 µm. (B) Quantification of fluorescence in images of A.

Figure 3B. Do you expect to see a shift of Hap43-Myc between high and low iron in WT as in figure 2B? Does this result also demonstrate the potential implication of another kinase for Hap43 phosphorylation as most of Hap43 proteins left the nucleus in iron replete condition in ssn3 deletion mutant?

We thank the reviewer for these pertinent comments. To visualize a clear mobility shift of phosphorylated Hap43-Myc protein in iron-replete medium compared to that under iron-depleted conditions (Figure 2B), we technically need to run the samples on SDS-PAGE gel for extremely long time because the protein size of Hap43-Myc is relatively big (~120 kDa). In Figure 3B, the changes of cellular localization of Hap43-Myc in WT and *ssn3Δ/Δ* mutant were analyzed and importantly, we still observed an apparent shift in the electrophoretic mobility of Hap43 in *ssn3Δ/Δ* mutant grown under high iron condition (left panel, lane 1 vs. 2; the shift was not as obvious as that in Figure 2B), although the protein samples were run on SDS-PAGE gel for a short period of time.

Previous studies have demonstrated that the classical Ras/cAMP/PKA signaling pathway is critically important for detecting environmental conditions and facilitating adaptation to nutrient availability in pathogenic fungi such as *Cryptococcus neoformans* and *Ustilago maydis*. This pathway has been found to be connected to regulation of ROS stress and iron homeostasis. We therefore considered the possible involvement of this pathway that may contribute to Hap43 phosphorylation. To test the possibility, we individually expressed Hap43-Myc in a list of *C. albicans* mutants lacking each of the key players of this pathway (*ras1Δ/Δ*, *gpa2Δ/Δ*, *cyr1Δ/Δ* and *tpk1 Δ/Δ*) and compared the levels of Hap43 when both WT and mutant cells were grown under iron-replete conditions. Unfortunately, we found that this pathway appears to be unnecessary for Hap43 phosphorylation and the kinase Tpk1 does not behave as Ssn3 (Author response image 2). We will seek to test more protein kinase candidates in our future studies.

In addition, we also attempted to identify the proteins binding to Hap43 by IP-mass spectrometry when *C. albicans* cells expressing Hap43-TAP were grown under iron-replete conditions. The trials turn out to be unrealistic since our gel staining results did not yield any specific proteins in Hap43-TAP samples compared to that in untagged WT (Author response image 2), more likely due to the fact that high iron triggered the protein degradation of Hap43-TAP resulting in the low levels of the bait protein. Future direction will be generating a strain overexpressing Hap43-TAP and redo the IP-mass spectrometry experiments.

On the other hand, we did visualize a band with relatively strong signal (~90kDa, almost the same size as Hap43-TAP) in Author response image 2. The band was cut and sent for phosphorylation site mapping by mass spectrometry. Unfortunately, our trial was failed and no phosphorylation sites were identified. We were told by the company technician that the amount of enriched protein was still insufficient for mapping. We are now enlarging the culturing volume and optimizing our purification protocol.

**Author response image 2. sa2fig2:** (A) Immunoblots of Hap43-Myc recovered from indicated *C. albicans* cells under iron-replete conditions. a-tubulin, internal standard. (B) Strain expressing Hap43-TAP was inoculated to YPD medium and grown overnight at 30^o^C for protein purification. Cell extracts were sequentially immunoprecipitated with lgG Sepharose and calmodulin Sepharose. Proteins were identified by gel electrophoresis separation and silver staining. The arrow represents Hap43.

References:

Choi J *et al.* (2015) The cAMP/protein kinase A signaling pathway in pathogenic basidiomycete fungi: connections with iron homeostasis. *J Microbiol.* 53(9): 579-87.

Martins TS *et al.* (2018) Signaling pathways governing iron homeostasis in budding yeast. *Mol Microbiol.* 109(4): 422-432.

Bouchez C and Devin A (2019) Mitochondrial biogenesis and mitochondrial reactive oxygen species (ROS): a complex relationship regulated by the cAMP/PKA signaling pathway. *Cells*. 8(4):287

Hogan DA and Sundstrom P (2009) The Ras/cAMP/PKA signaling pathway and virulence in *Candida albicans*. *Future Microbiol*. 4(10): 1263-70.

L254-256. At this point we can only say that Hap43 size shift and cellular localization can be modulated by SSN3. We do not even know whether Hap43 can be phosphorylated by SSN3, which may require purified proteins to prove.

Yes, there is no direct evidence that the protein kinase Ssn3 mediates Hap43 phosphorylation when *C. albicans* cells were grown under high iron conditions, which requires an in vitro protein kinase assay using purified recombinant proteins. We agree with the reviewer’s comment that the protein kinase Ssn3 acts to modulate the phosphorylation-mediated mobility shift and cellular localization of Hap43. The descriptions have been rephrased in the revised manuscript. (Line 204 on page 11; Lines 231-232 on page 12; Line 242 on page 13; Line 247 on page 13; Line 315 on page 16; Line 592 on page 30; Line 1160 on page 55; Line 1181 on page 57)

Our data suggest that the protein kinase Ssn3 plays a key role in modulating Hap43 phosphorylation, mainly based on the following evidence: (1) Hap43 phosphorylation could be heavily affected by Ssn3, as we observed that *C. albicans* mutant lacking *SSN3* (*ssn3Δ/Δ*) or expressing a predicted kinase-dead allele of Ssn3 (Ssn3^D325A^) showed the abolishment of increased mobility and consequently, the level of Hap43 is comparable with that found under iron-depleted conditions (Figure 2E); (2) Ssn3 physically interacts with Hap43 (Figure 2F). Actually, one direction of our ongoing follow-up study is to confirm whether Ssn3, or other kinases, is the kinase who directly mediates the phosphorylation of Hap43. The experiments will include the in vitro kinase assay and candidate screening using kinase null mutants.

L256-258. Current evidence only supports that Hap43 expression and localization depends on SSN3 expression.

We accepted the reviewer’s comment and rephrased the description in the revised manuscript. (Line 258-260 on page 14)

L275-276. There is no data supporting the dependence of Hap43 phosphorylation by SSN3. Hap43 phosphorylation have never been demonstrated.

We accepted the reviewer’s comment and rephrased the conclusion in the revised manuscript. (Line 204 on page 11; Lines 231-232 on page 12; Line 242 on page 13; Line 247 on page 13; Line 315 on page 16; Line 592 on page 30; Line 1160 on page 55; Line 1181 on page 57)

Figure 3—figure supplement 1. Do you expect that the dynamic of Hap43-Myc degradation was much similar to the condition with MG132 in figure 3E?

We apologize for the lack of clarity. Studies have shown that in eukaryotic cells, lysosomal proteolysis and the ubiquitin-proteasome system represent two major protein degradation pathways mediating protein degradation (Lecker et al., 2006). To clarify the exact proteolytic pathway implicated in Hap43 turnover under iron-replete conditions, we performed two experiments by incubating cells with specific and selective inhibitors of the lysosome (Chloroquine) or the proteasome (MG132). The results from both Figure 3E and Figure 3—figure supplement 1 strongly support that the phosphorylated form of Hap43 is prone to be degraded through the proteasomal pathway but not lysosomal pathway, as we found that inhibition of the proteasomal pathway by MG132 treatment abrogates the degradation of Hap43 (Figure 3E), but it is not the case when the lysosomal pathway is blocked by chloroquine (Figure 3—figure supplement 1). So, the dynamic of Hap43-Myc degradation should not be the same when two different treatments were applied.

Figure 5C. Is it the time after Hap43-Myc overexpression? Better to describe more clearly in the figure legends.

We thank the reviewer for pointing out this lack of clarity. As described in the revised figure legends, overexpression of Hap43-Myc was induced under the control of a doxycycline (DOX) inducible promoter (TetO-Hap43-Myc). Exponential-phase cells grown in iron-replete (YPD) medium supplemented with 50 mg/ml doxycycline were harvested, washed and re-suspended in fresh YPD medium only, YPD supplemented with 200 mM FeCl_3_, a combination of 200 mM FeCl_3_ and 20 mM N-acetyl-L-cysteine (NAC) (C) or 20 mM menadione for 120 min (D). (Lines 1266-1270 on page 64)

Figure 5D. Why did you have Hap43-Myc expression 120 min after induction here, but not in figure 4C in YPD only culture condition?

We apologize for the lack of clarity. The experiments in Figure 4C and 5D were conducted in completely different conditions. In Figure 5D, Hap43-Myc was expressed under the control of a doxycycline (DOX) inducible promoter. In Figure 4C, Hap43-TAP was expressed under the control of its own promoter.

L409. The current data do not demonstrate the ubiquitination-dependent degradation Hap43, but just show SSN3-dependent ubiquitination (Figure 3F) and phosphorylation-dependent degradation (Figure 4F), respectively.

The description has been changed in the revised manuscript. (Line 418 on page 21)

L478-483. No further data to support the conclusion. I would suggest to move this part to discussion.

Thank the reviewer’s for this pertinent comment. We followed the reviewer’s suggestion and moved this part to the Discussion section. (Lines 632-639 on page 32)

Reviewer #2 (Recommendations for the authors):The authors present a significant body of work that demonstrates a novel mechanism by which the human commensal yeast Candida albicans can adapt to high levels of iron in the GI tract. The paper includes a detailed set of carefully conducted experiments in vitro and in vivo that show a mechanisms that links high iron levels to the management of reactive oxygen species levels in C. albicans cells. This mechanism provides a compelling argument for a strategy that will allow colonisation of the GI tract. The mechanism also represents a potential route to manage C. albicans colonisation of the GI tract, an issue that is increasingly linked to a number of gut disorders.This is an excellent study that presents a novel mechanism that allows C. albicans to thrive in an iron rich environment such as the GI tract. It is an elegant study and I had very few issues with the data presented, which were compelling, well delivered and of very high quality. I do not believe that additional experiments are required as the paper does lead a significant advance in the field.

We are highly thankful to the reviewer for encouragement and positive affirmation given to our work reported in this manuscript.

Some points to consider that I think would improve the paper are as follows:Abstract – some grammatical errors are present, please review.

We are very grateful for the reviewer’s comments on the manuscript. According to your advice, we have revised the abstract. (Lines 19-30 on page 2)

Importance – "iron homeostasis is critical for creatures"?? this does not read as a scientific statement, please adjust.

Thank the reviewer for the valuable suggestion. We have revised our description accordingly. (Line 32 on page 3)

The importance summary requires revision to correct a number of many grammatical errors. It also does not really convey the importance of this research, why is it of interest?

Thank you very much for your valuable comments. Based on your comments, we have made extensive revisions in this session. (Lines 32-44 on page 3)

Results – figure 1 – line 164 change "neglectable" to negligible.

We have corrected it as suggested. (Line 167 on page 9)

The adjustment of diet to 400mg/Kg to ensure that Ros levels were increased needs to be properly discussed, as in a normal diet HAP43 was not required, the results are therefore relevant to either poor diet, a gut disease associated with high ROS or transient increases in iron associated, or is it that Hap43 is required to buffer transient increases in iron level to aid commensalism? Please include a discussion of this.

We are highly thankful to the reviewer for the thought-provoking suggestions. We carefully went through the valuable information provided by the reviewer and revised our discussion. (Lines 609-617 on page 31)

Figure 6 It was curious that TRR1 levels were not altered despite binding showing a significant increase, this should be discussed.

We thank the reviewer for raising this point. In some cases, we did not observe significant changes in expression of several indicated antioxidant genes such as *TRR1* and *GSH1*, even though we are aware of a highly significant increase in Hap43 binding to their promoters based on our ChIP assays, arguing the involvement of other regulators, such as Cap1 and Tsa1/Tsa1B, in modulating the transcript levels of these genes, in addition to Hap43. (Lines 487-491 on page 25)

References:

Urban C *et al.* (2005) The moonlighting protein Tsa1p is implicated in oxidative stress response and in cell wall biogenesis in *Candida albicans*. *Mol Microbiol*, *57*(5), 1318-1341.

Wang Y *et al.* (2006) Cap1p is involved in multiple pathways of oxidative stress response in *Candida albicans*. *Free Radic Biol Med*, *40*(7), 1201-1209.

The discussion itself is rather brief, it would be useful to discuss how the iron may be absorbed by Candida cells as it does not have, as yet, identified siderophores. There is evidence for several iron uptake mechanisms and this should be mentioned. Although I do not suggest carrying out more experiments to expand the mechanism, presumably identifying the upstream mechanism would be the next step? and this should be discussed. It would also be important to discuss this mechanism within the context of the GI tract, which presumably has highly variable levels of iron. Is the system a buffer to ensure that the cells can remain competitive in the fluctuating environment of the GI tract for example? how would the system interplay with variable oxygen levels found in the GI tract?

We sincerely appreciate these valuable comments and suggestions, which helped up to improve the quality of the manuscript. We have rewritten the Discussion by integrating all these priceless advices from the reviewers. (Lines 513-572 on pages 26-29)

Reviewer #3 (Recommendations for the authors):In this work, the authors show a role in gut colonization for the well-established regulator of Candida albicans iron homeostasis, Hap43. They link an increased ability of a Hap43 deletion mutant to colonize the gut of mice under a high-iron diet to the repressor role of Hap43 in oxidative stress response (and the iron-dependent Fenton reaction that creates such stress). They go on to show the interplay between posttranslational modification, localization, degradation and gene regulation by Hap43 in response to shifting iron levels.Overall, there are some very interesting findings in this manuscript, and it definitely adds to our understanding of iron-dependent regulation in C. albicans. The set of experiments is rather comprehensive and supports the conclusions. It will certainly be of interest to scientists in the field of Candida and fungal pathogenesis in general, as well as those interested in the role of micronutrients in host-pathogen interactions.Experiment-wise, I was fully convinced by the authors' systematic approach, and all the conclusions seem to be fully supported by the data.

Thank you very much for your affirmation of the content of our paper.

My main concern is that some aspects are not as novel as depicted in the text (and the relevant literature is missing from the cited references). For example, the regulation of the oxidative stress response in dependence on iron (and especially Hap43) has been described in Chakravarti et al. (2016). This is implied, but not stated, in the beginning (around line 390), there citing a review instead. However, later (line 550) the text states that "… we discovered the role of the transcription factor Hap43 in modulation of the transcription of antioxidant genes in response to iron". I believe this connection has been known before?

We agree with the reviewer’s critical points. Indeed, a previous study has identified that the CCAAT-binding complex (with the Hap31 or Hap32 subunits, with or without Hap43) is essential for the iron-dependent regulation of the oxidative stress response in *C. albicans* (Chakravarti *et al.*, 2017). We sincerely apologize for not citing their work and the related descriptions have been corrected in the revised manuscript. (Lines 625-629 on pages 31-32)

Reference:

Chakravarti A *et al.* (2017). The Iron-Dependent Regulation of the *Candida albicans* Oxidative Stress Response by the CCAAT-Binding Factor. *PLoS One*, 12(1), e0170649.

Similarly, while it was interesting to see multiple phosphorylation sites in Hap43, the observation of phosphorylation and subsequent degradation of Hap43 was not that surprising. Essentially the same has been shown before for its A. fumigatus homologue, HapX, complete with the role of the proteasome in this process (in the similarly uncited Lopez-Berges et al. (2021)). In the present manuscript, the authors show much more data, including the localization and the responsible kinase, but the data should be put into context of the Aspergillus HapX findings (e.g. with the non-phosphorylatable mutant tested there or the identical effect of MG132).

Again, we sincerely apologize for not citing the important findings discovered in *Aspergillus fumigatus*, which are summarized as follows. (1) Similar to Hap43, the *A. fumigatus* homologue HapX also undergoes post-translational modifications (ubiquitination, sumoylation and phosphorylation) during iron-replete conditions and these post-translational regulation is important to control iron homeostasis in *A. fumigatus* (Lopez-Berges *et al.*, 2021); (2) *A. fumigatus* HapX exhibits inverse regulatory activities by mediating repression or activation of vacuolar iron storage depending on the ambient iron availability, suggesting that HapX appears to be important for regulating both iron resistance and adaptation to iron starvation (Gsaller *et al.*, 2014); and (3) Interestingly, *C. albicans* Hap43, in stark contrast to HapX of *A. fumigatus*, appears to play only a minor role in mediating the adaption to iron excess, although this factor did act positively for the activation of iron uptake genes during iron starvation (Skrahina *et al.*, 2017).

We have added these supporting references in the revised manuscript. Moreover, we followed the reviewer’s comments and combined our results with the related observations from these studies. (Lines 315-327 on pages 16-17)

References:

Lopez-Berges MS *et al.* (2021) The bZIP Transcription Factor HapX Is Post-Translationally Regulated to Control Iron Homeostasis in *Aspergillus fumigatus*. *Int J Mol Sci*, 22(14).

Gsaller F *et al.* (2014) The Janus transcription factor HapX controls fungal adaptation to both iron starvation and iron excess. *EMBO J*, 33(19), 2261-2276.

Skrahina V *et al.* (2017) *Candida albicans* Hap43 Domains Are Required under Iron Starvation but Not Excess. *Front Microbiol*, 8, 2388.

The connection to the diet is interesting, although a bit more critical discussion may be in order. How realistic are these conditions in the human colonization situation -- the Fe input is higher than the highest concentration given for human feces in the introduction? And why did the same effect of the Hap43 mutant occurs in the *Drosophila* model, although it does not seem to be especially iron-rich?

Studies have shown that adult iron intake is 10-15 mg per day, a pregnant woman should receive a daily iron up to 30-60 mg, and for the patients who develop with iron deficiency anemia, the recommendation is to consume an additional 60-90 mg of iron/ day (for a total of 120 mg per day). In our study, the mice fed a high iron diet (HFD) receive a daily iron supplement of 2-4 mg, lower than the daily dose consumed by human.

As for the *Drosophila* model, we checked the publications and did not find the information supporting that the gut of flies is iron-rich and has high levels of ROS. Actually the reviewer #1 had the same concern. We therefore accepted the reviewers’ comments and removed the *Drosophila* parts in the revised manuscript, in order to avoid possible misleading information.

References:

Piskin E *et al.* (2022) Iron Absorption: Factors, Limitations, and Improvement Methods. *ACS Omega* 7(24): 20441-20456.

Liu K *et al.* (2012) Iron deficiency anaemia: a review of diagnosis, investigation and management. *Eur J Gastroenterol Hepatol.* 24(2): 109-16.

Duarte AFM *et al.* (2021) Oral iron supplementation in pregnancy: current recommendations and evidence-based medicine. *Rev Bras Ginecol Obstet* 43(10): 782-788

Haram K *et al.* (2001) Iron supplementation in pregnancy – evidence and controversies. *Acta Obstet Gynecol Scand.* 80(8): 683-8.

Ortiz R *et al.* (2011) Efficacy and safety of oral iron(III) polymaltose complex versus ferrous sulfate in pregnant women with iron-deficiency anemia: a multicenter, randomized, controlled study. *J Matern Fetal Neonatal Med*. 24(11): 1347-52

Lee JK *et al.* (2021) Dietary iron intake in excess of requirements impairs intestinal copper absorption in Sprague Dawley Rat Dams, causing copper deficiency in suckling pups. *Biomedicines*. 9(4): 338.

The sections on importance (and to some extent, the abstract) are in need of some proofreading/rewriting -- e.g. "in the GI tract where is iron-replete environment", "the mystery in iron-replete conditions of Hap43 has never been uncovered" and so on. This is strikingly different to the rest of the text, which is quite easy to follow and rather well-written.

We are grateful to the reviewer for these constructive suggestions. The sections on both Importance and Abstract have been rewritten according to the reviewer’s advices. (Lines 32-44 on page 3)